# Beyond Object Recognition: A New Benchmark towards Object Concept Learning

## Abstract

Understanding objects is a central building block of artificial intelligence, especially for embodied AI. Even though object recognition excels with deep learning, current machines still struggle to learn higher-level knowledge, e.g., what attributes does an object have, what can we do with an object. In this work, we propose a challenging **Object Concept Learning** (OCL) task to push the envelope of object understanding. It requires machines to reason out object affordances and simultaneously give the reason: *what attributes make an object possesses these affordances*. To support OCL, we build a *densely* annotated knowledge base including extensive labels for three levels of object concept: categories, attributes, and affordances, and their causal relations. By analyzing the causal structure of OCL, we present a baseline, Object Concept Reasoning Network (OCRN). It leverages causal intervention and concept instantiation to infer the three levels following their causal relations. In experiments, OCRN effectively infers the object knowledge while follows the causalities well. Data and code will be publicly available.

## 1 Introduction

Object understanding is essential for intelligent machines. Recently, benefited by deep learning and large-scale datasets (Deng et al., 2009; Lin et al., 2014), category recognition (Krizhevsky et al., 2012; Ren et al., 2015) has made tremendous leaps of progress. But to close the gap between human and machine perception, machines need to pursue deeper object understanding, e.g., recognizing higher-level attributes (Isola et al., 2015) and affordances (Gibson, 2014), which may help them establish the concept of objects (Martin, 2007) when interacting with complex contexts.

Category `apple` is a symbol indicating its referent (real apple). In line with symbol grounding (Harnad, 1990), machines should learn knowledge beyond category to approach concept understanding. According to cognition studies (Ross, 2008; Martin, 2007), attributes depicting objects from the physical/visual side play an important role in object understanding. Thus, many works (Lampert et al., 2009; Xiao et al., 2010; Farhadi et al., 2009) studied to ground objects with attributes, e.g., a `hammer` consists of a `long` handle and a `heavy` head. Moreover, attributes can depict object *state* (Isola et al., 2015). An elegant characteristic of attributes is cross-category: objects of the same category can have various states (`big` or `fresh apple`), whilst various objects can have the same state (`sliced orange` or `carrot`). If the category is the **first** level of object concept, the attribute can be seen as the **second** level closer to the physical fact.

However, recognizing attributes is still far away from concept understanding. Given a `hammer`, we need to know it can be `held` to `hit` nails. That is, requiring machines to infer affordance (Gibson, 2014) indicating what actions humans can perform upon objects. Here, we refer to affordance as the **third** level, which is closely related to common sense and causal inference (Gibson, 2014). Although affordance learning has been studied in robotics (Do et al., 2018; Hermans et al., 2011) and vision (Chuang et al., 2018; Zhu et al., 2014) communities in decades, it is still challenging. First, previous works (Nguyen et al., 2017; Fouhey et al., 2015) often focus on recognizing affordance solely. But we usually infer affordance based on attribute observation. When we need to knock in a nail without a `hammer` at hand, we may find other `hard` or `heavy` objects instead, e.g., a `thick` book. This profoundly reveals the **causality** between attribute and affordance. Second, previous works are designed for category/scale/scene-limited tasks, e.g., in Zhu et al. (2014), 40 objects and 14 affordances are included; Hermans et al. (2011) collect 375 indoor images of 6 objects, 21

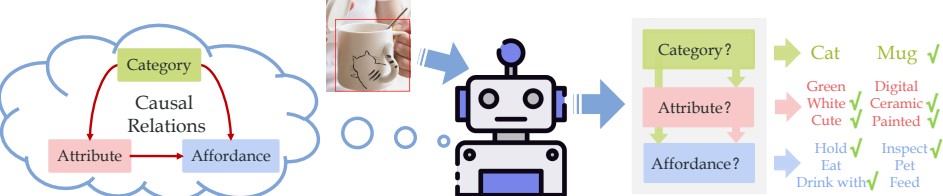

Figure 1: For intelligent and embodied agents, understanding the daily objects requires the ability to perceive not only object category but also attributes and affordance. Thus, in **OCL**, we try to reveal object concept learning in both three levels and explore their profound causal relations.

attributes, and 7 affordances; a recent dataset (Nguyen et al., 2017) contains 10 indoor objects and 9 affordances. Hence, they cannot support general affordance reasoning for large-scale applications.

To reshape object learning, we believe it is essential to look at the above three levels in a **unified** and **causal** way based on an extensive knowledge base. Hence, we move a step forward to propose the object concept learning (OCL) task: given an object, machines need to infer its category, attributes, and further answer "*what can we do upon it and why*". In a nutshell, machines need to reason affordance based on object appearance, category, and attributes. To this end, we build a large-scale and dense dataset consisting of **381** object categories, **114** attributes, and **170** affordances. It contains **80,463** images of diverse scenes, **185,941** object instances in different states. Different from previous works (Chao et al., 2015; Hermans et al., 2011; Zhu et al., 2014), OCL offers a more nuanced angle. It includes: (1) **category**-level attribute ($A$) and affordance ($B$) labels; (2) **instance**-level attribute ($\alpha$) and affordance ($\beta$) labels. Besides, we consider and annotate the *causal relations* between three levels to evaluate the reasoning ability of models and keep the follow-up methods from fitting data only. Accordingly, based on the causal structure of OCL, we also propose a *neuro-causal* method, **O**bject **C**oncept **R**easoning **N**etwork (**OCRN**), as the future baseline. It operates the *instantiation* from category-level to instance-level, and leverages *causal intervention* (Pearl et al., 2016) to alleviate the bias from categories and infer attributes and affordances. OCRN outperforms a host of baselines and shows impressive performance while follows the causal relations well.

In summary, our contributions are threefold: (1) Introducing the object concept learning (OCL) task poses challenges and opportunities for deeper object understanding and knowledge-based reasoning. (2) Building a dataset consists of diverse objects, elaborate attributes and affordances, together with their causal relations to benchmark OCL. (3) An object concept reasoning network (OCRN) is introduced to reason three levels with concept instantiation and debiasing, which performs well.

## 2 RELATED WORK

**Object Attribute.** Attribute depicts the visual/physical properties like color, size, shape, etc. It usually plays the role of intermedia between pixels and higher-level concepts, e.g., prompting object recognition (Farhadi et al., 2009), affordance learning (Hermans et al., 2011), zero-shot learning (Lampert et al., 2009), and object detection (Kumar Singh et al., 2018). Recently, several large-scale datasets (Farhadi et al., 2009; Xiao et al., 2010; Liu et al., 2017; Patterson & Hays, 2016; Isola et al., 2015; Krishna et al., 2016; Hudson & Manning, 2019) are released. For attribute recognition, recent works have made progress. Beside direct multi-label classification (Lampert et al., 2009; Parikh & Grauman, 2011; Xiao et al., 2010; Patterson & Hays, 2016) or leveraging the correlation between attribute-attribute and attribute-object (Hwang et al., 2011; Chen & Grauman, 2014; Mahajan et al., 2011), intrinsic properties (compositionality, contextuality (Misra et al., 2017; Nagarajan & Grauman, 2018), symmetry (Li et al., 2020b)) of attribute-object are also proven useful.

**Object Affordance.** The concept of affordance is introduced by Gibson (2014). Affordance learning has two canonical paradigms: direct mapping (Fouhey et al., 2015) or indirect method (Zhu et al., 2014; Zhao & Zhu, 2013; Wang et al., 2017; Roy & Todorovic, 2016) with intermedia like object category, attribute, and 3D contents. Some studies also tried to learn affordance from human-object interactions, i.e., encoding the relationship between object and action (Gupta & Davis, 2007; Yao et al., 2013; Kato et al., 2018). Visual Genome (Krishna et al., 2016) provides visual relationships between objects in the scene graph, including some human actions instead of affordances. However, these relationships cover limited and sparse affordances. Differently, we use easily accessible object images as the knowledge source, and *densely* annotate all attributes/affordances for all object instances. Besides the vision community, the robot community also pays much attention to affordance (Do et al., 2018; Pinto & Gupta, 2016; Thermos et al., 2017; Pinto et al., 2016), especially for

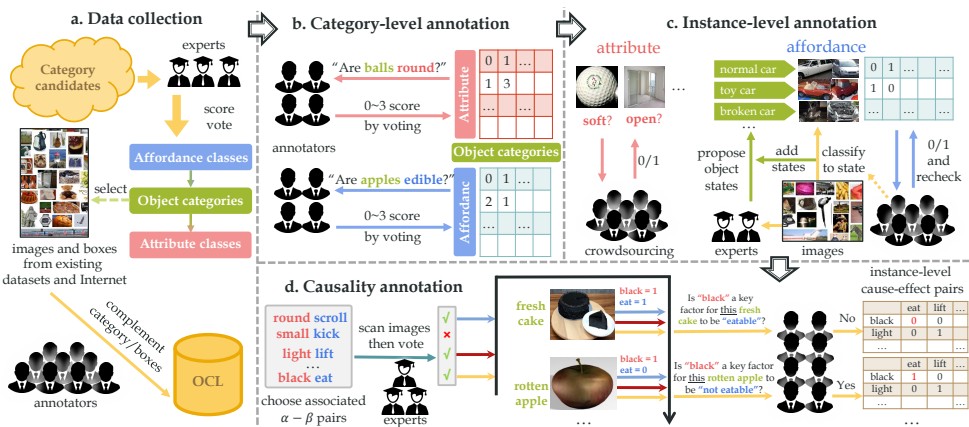

Figure 2: OCL construction. a) Data collection. b) Annotating category-level attributes and affordances. c) Annotating instance-level attributes and affordances. d) Finding direct causal relations.

object grasping and manipulation. Recently, several datasets (Nguyen et al., 2017; Zhu et al., 2014; Chao et al., 2015; Chuang et al., 2018) have been proposed. Zhu et al. (2014) built a knowledge base containing attribute, affordance, human pose, and human-object spatial configuration. But labeling pose and human-object relative location are costly. Chao et al. (2015) proposed a *semantic* category-level affordance dataset including 91 COCO objects (Lin et al., 2014) and 957 affordances.

**Causal Inference.** There is increasing literature on exploiting causal inference (Pearl et al., 2016) in machine learning, especially with causal graphical models (Spirtes et al., 2000; Pearl et al., 2016), including feature selection (Guyon et al., 2007) and learning (Chalupka et al., 2014), video analysis (Pickup et al., 2014; Lebeda et al., 2015), reinforcement learning (Nair et al., 2019; Dasgupta et al., 2019), etc. Recently, Wang et al. (2020a) studied the causal relation between objects in images and used intervention (Pearl et al., 2016) to alleviate the observation bias in detection. Atzmon et al. (2020) analyze the causal generative model of compositional zero-shot learning and disentangle the representations of attributes and objects. In this work, we explore the causal relations between three object levels and apply backdoor adjustment (Pearl et al., 2016) to alleviate the existing bias.

## 3  KNOWLEDGE BASE CONSTRUCTION

Our goal is to construct a dataset that can characterize abundant knowledge in object category, attribute, affordance, and causal relations. The construction process is given as follows (Fig. 2).

**Data Collection. (1) Affordance**: We collect 170 affordances out of 1,006 candidates from widely-used action/affordance datasets (Chao et al., 2015; Gu et al., 2018; Chao et al., 2018; Gupta & Malik, 2015; Zhu et al., 2014; Nguyen et al., 2017) in view of generality and commonness. **(2) Category**: Considering the taxonomy (WordNet (Fellbaum, 2012)) and diversity, we collect 381 objects out of 1,742 candidates from object datasets (Farhadi et al., 2009; Xiao et al., 2010; Patterson & Hays, 2016; Liu et al., 2017; Lin et al., 2014). **(3) Attribute**: When viewed through the lens of the causal relation between attribute-affordance, we manually filter 500 most frequent attributes from large-scale attribute datasets (Farhadi et al., 2009; Xiao et al., 2010; Patterson & Hays, 2016; Liu et al., 2017; Krishna et al., 2016) and choose 114 attributes, covering colors, deformations, supercategories, various surface, geometrical, and physical properties. **(4) Image**: We extract 75,578 images from object datasets (Farhadi et al., 2009; Xiao et al., 2010; Patterson & Hays, 2016; Liu et al., 2017; Krishna et al., 2016), together with their Ground Truth (GT) boxes. To ensure diversity, we further manually collected 4,885 Internet images of selected categories. Then, we annotate the missing box and category labels for all instances via crowdsourcing. Finally, **185,941** instances of **381** categories from **80,463** images are collected, i.e., average 488 instances per category and 2.31 boxes per image. More details and class lists are in *Appendix Sec. A.11*. OCL is long-tail distributed, where the head categories have over 5,000 instances each, but the rarest categories have only 9 instances, which challenges the robustness of vision systems.

**Attribute Annotation.** We offer annotations in two levels: (1) **Category-level attribute** (A) contains common sense. For each *category*, we annotate its **most common** attributes. In concept learning, the usage of category-level annotation as common knowledge can date back to Osherson et al. (1991). Following Osherson et al. (1991), to avoid bias, annotators are given *category-attribute*

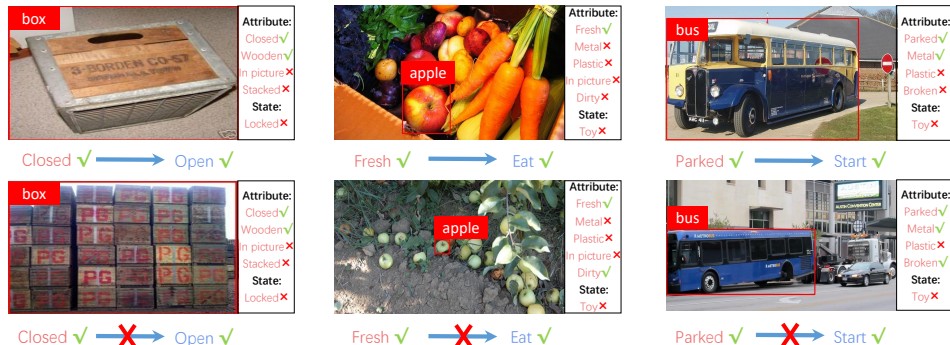

Figure 3: OCL examples. Causal relations between $\alpha$ (red), $\beta$ (blue) in various contexts are listed.

| Dataset | # Image | # Instance | # Object | # Attribute | # Affordance |
|---|---|---|---|---|---|
| APY (Farhadi et al., 2009) | 15,339 | 15,339 | 32 | 64 | / |
| SUN (Xiao et al., 2010) | 14,340 | 14,340 | 717 | 102 | / |
| COCO-a (Patterson & Hays, 2016) | 84,044 | 188,426 | 29 | 196 | / |
| ImageNet150k (Liu et al., 2017) | 150,000 | 150,000 | 1,000 | 25 | / |
| Chao et al. (2015) | / | / | 91 | / | 957 ($B$) |
| Hermans et al. (2011) | 375 | - | 6 | 21 | 7 |
| Zhu et al. (2014) | 4,000 | 4,000 | 40 | 57 | 14 |
| OCL | 80,463 | 185,941 | 381 | 114 | 170 |

Table 1: Comparison of *dense annotated* datasets. OCL provides the category- and instance- level attributes ($A$, $\alpha$), affordances ($B$, $\beta$).

*pairs* (category *names*, not images) and multi-annotator vote to build the binary category-level attribute matrix $M_A$. ([381, 114]). (2) **Instance-level attribute** ($\alpha$) is the individual attributes of **each instance**. The annotation unit is an *attribute-instance pair*. Each pair is labeled by multi-annotator.

**Affordance Annotation.** There are also two levels of granularity: (1) **Category-level affordance** $B$, similar to $A$, is annotated in *category-affordance pairs*, indicating the common affordances of each category. Following Chao et al. (2015), the annotators label category-level affordance matrix $M_B$ ([381, 170]) similar to $A$ annotation. (2) **Instance-level affordance** $\beta$ is annotated for **every instance** with the help of *object states* (Isola et al., 2015). As $B$ is determined by common states, objects in specific states may have different affordances from $B$, e.g., we cannot board a flying plane. Moreover, if a service robot finds a broken cup, it may infer that the cup can still hold water as it is trained with $B$ labels. Thus, we need detailed $\beta$ beyond $B$. As the instances in the same state should have similar $\beta$ (all rotten apples cannot be eaten), six experts first conclude the states. In total, **1,376** states are defined, and each category has 3.6 states on average. Next, $\beta$ is annotated for each state and the instances are assigned with *beta* according to their states. More details please refer to *Appendix Sec. A.2* It is worth noting that, as states seem to be a beneficial latent variable for OCL, in practice, we do not use it in models because the states we use are a *subset* of whole state space, and there can be *unseen* states in real-world data.

**Causal Relation Annotation.** We annotate *instance*-level (considering the context of each instance) causality to answer *which attribute(s) are the critical and direct causes of a certain affordance?* **(1) Filtering** to avoid the combinatorial explosion. Initially, we need to make binary decisions on all *instance-$\alpha$-$\beta$* triplets, which is far beyond handleable. Fortunately, we find that **most** $\alpha$-$\beta$ classes (e.g., shiny and kick) are meaningless and always of no causality on any instances. Thus, we can firstly exclude the most impossible pairs, meanwhile, guarantee the completeness of causality. Finally, we obtain about 10% $\alpha$-$\beta$ classes as candidates. **(2) Instance-level causality**: we also adopt object states as a reference. Multiple annotators have been involved for each *state-$\alpha$-$\beta$* triplet and are asked whether the specific attribute is the *direct* cause of this affordance in this state. The answers are combined and checked for all instances of a state. Finally, we obtain about 2 M *instance-$\alpha$-$\beta$* triplets of causal relations. As we have labelled all $\alpha$ and $\beta$ for all instances, the causal relations would be in four situations: [0,0], [1,1]; [0,1], [1,0]. The former two are "positive", e.g., fresh(1/0)$\rightarrow$eat(1/0) for an apple. While the last two are "negative", e.g., broken(1/0)$\rightarrow$drive(0/1) for a *car*.

**Discussion.** Fig. 3 shows some examples of OCL. These clear causalities are not thoroughly studied in previous datasets (Zhu et al., 2014; Nguyen et al., 2017; Fouhey et al., 2015; Hermans et al., 2011). We compare OCL with previous dense datasets in Tab. 1. More analysis figures and tables are in *Appendix Sec. A.3*.

# 4 OBJECT CONCEPT LEARNING

## 4.1 TASK OVERVIEW

Given an instance $I$ (image region in box $b_o$), OCL aims to infer attribute $\alpha$ and affordance $\beta$ while following the causalities. Formally, OCL can be described as:

$$< P_\alpha, P_\beta >= \mathcal{F}(I, P(O|I)), \Delta P_\beta = TDE[\mathcal{F}(I, P(O|I))], \quad (1)$$

where $P_\alpha, P_\beta$ are the probabilities of $\alpha, \beta$, given an object instance $I$ and its predicted category probability $P(O|I)$. $\Delta P_\beta$ means the Total Direct Effect ($TDE[\cdot]$) (Pearl et al., 2016) of affordance prediction change after we operate upon a model $\mathcal{F}(\cdot)$. $\Delta P_\beta$ is expected to follow the GT causal relation between attribute-affordance. We will detail the reasoning evaluation in Sec. 5.

We spilt images into train, validation (val) and test sets with 56,916:14,446:9,101 images. The val and test sets cover 221 of the 381 categories, and the train set covers all categories. OCL is a long-tailed recognition task and requires generalization to cover the whole object category-attribute-affordance space with imbalanced information. Thus, it is challenging for current vision systems without the reasoning ability to understand the causalities.

## 4.2 OBJECT CONCEPT REASONING NETWORK

Here, we first analyze the causal structure of OCL and then propose a basic system, Object Concept Reasoning Network (OCRN), as the future baseline for the OCL task.

**OCL Causal Graph.** We use the causal graph to shed light on the subtle causalities of OCL in Fig. 4. Causal graph (Pearl et al., 2016) indicates the underlying causalities between variables $O, A, B; I, \alpha, \beta$. Directed edges represent the causal directions. According to the prior knowledge about the causalities between three levels, a hierarchical structure is depicted: **(a)** the **inner** triangle with dotted lines is the **category**-level: object category $O$ decides the category-level attributes $A$ and affordances $B$;

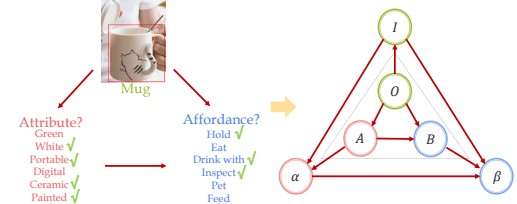

(a) **OCL**: Object Concept Learning     (b) Causal Graph

Figure 4: OCL causal graph. $I$ is object appearance. $I, \alpha, \beta$ are the **instantiations** of $O, A, B$.

**(b)** the **outer** triangle is the **instance**-level: instance visual appearance $I$, instance-level attributes $\alpha$ and affordances $\beta$. As $I$ indicates the *physical noumenon*, it is the source of semantic and functional properties and decides $\alpha, \beta$. As attributes can affect affordances, in both two levels, there are edges from attributes to affordances. From another perspective, $I, \alpha, \beta$ are the *instantiations* of $O, A, B$ respectively. We can use a whole node $O'$ to represent $O, A, B$ (Fig. 5).

**Object Category Bias.** Formally, OCL is depicted as $P(\alpha|I)$ and $P(\beta|I, \alpha)$. Following the causal graph, we represent nodes $\{I, A, B, \alpha, \beta\}$ as $\{f_I, f_A, f_B, f_\alpha, f_\beta\}$ respectively in latent space. $f_I$ is the RoI pooling feature of $b_o$ extracted by a COCO pre-trained ResNet-50 (He et al., 2016). As the samples of different object categories are usually imbalanced, conventional methods may suffer from severe *category bias* (Wang et al., 2020a). For example, in OCL, animal accounts for 22% instances and home appliance only accounts for 3%. In $P(\alpha|I)$, category bias is imported following

$$P(\alpha|I) = \sum_i^m P(\alpha|I, O_i)P(O_i|I), \quad (2)$$

where $P(O_i|I)$ is the predicted category probability. That is, $O$ is a confounder (Pearl et al., 2016) and pollutes attribute inference, especially for the *rare* categories (analyzed in Fig. 7, 8).

**Causal Intervention.** To tackle this problem, we propose OCRN utilizing intervention (Pearl et al., 2016) to deconfound the confounder $O$ for $\alpha$ (Fig. 5). That said, in $\alpha$ estimation, we use $do(\cdot)$ operation (Pearl et al., 2016) to eliminate the edge from $O$ to $I$, i.e., $P(\alpha|do(I))$ is

$$\sum_i^m P(\alpha|I, O_i)P(O_i) = \sum_i^m P(O_i)\sum_j^m P(\alpha|I, A_j)P(A_j|O_i) = \sum_i^m P(\alpha|I, A_i)P(O_i), \quad (3)$$

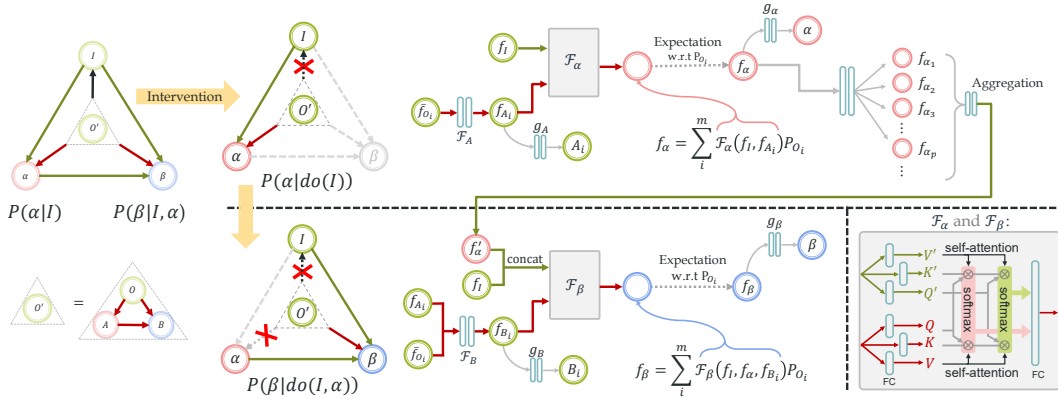

Figure 5: The overview of OCRN. The edge from $O$ to $I$ is deconfounded. Thus we can eliminate the bias from the $O$ imbalance. Equations below the graphs are the estimations of $\alpha, \beta$ w/ or w/o $do(\cdot)$. Attribute and affordance modules are the **instantiations** of category-level features. Mean object category feature $\bar{f}_{O_i}$ is converted to $f_{A_i}$ or $f_{B_i}$ by an FC, and then instantiated via $\mathcal{F}_\alpha(f_I, f_{A_i})$ or $\mathcal{F}_\beta(f_I, f_\alpha, f_{B_i})$. Causal paths marked in green are the conditions of $\mathcal{F}_\alpha(\cdot)$ or $\mathcal{F}_\beta(\cdot)$ implemented via attention policy. $f_\alpha$ and $f_\beta$ after intervention are the expectations of instantiated $\mathcal{F}_\alpha(f_I, f_{A_i})$ and $\mathcal{F}_\beta(f_I, f_\alpha, f_{B_i})$ w.r.t **prior** $P_{O_i}$. Linear-Sigmoid classifiers give the final predictions.

where $m = 381$. $A_j$ is the category-attribute vector of $j^{th}$ category. As $A$ is decided by $O$, $P(A_j|O_i) = 1$ iif $i = j$ and $P(A_j|O_i) = 0$ if $i \neq j$, where $O_i$ is the $i^{th}$ category and $A_j$ is the category-attribute of $j^{th}$ category. $P(O_i)$ is the **prior** probability of the $i$-th category (frequency in the OCL train set).

### 4.2.1 ATTRIBUTE ESTIMATION

To implement Eq. 3, we use *mean* object category feature to represent category-level attribute $A$. We take the mean feature ($\bar{f}_{O_i}$) of all instances in one category as the representation of the category ($O_i$). It is then mapped to attribute feature space: $f_{A_i} = \mathcal{F}_A(\bar{f}_{O_i})$, where $\mathcal{F}_A(\cdot)$ is a fully-connected layer (FC) (Fig. 5). The representation set of $A$ is $f_A = \{f_{A_i}\}_{i=1}^m$. For each category, we use $f_I$ of its all instances (train set) to calculate $\bar{f}_{O_i}$ in training and inference.

**Attribute Instantiation.** Next, we obtain $\alpha$ representation according to Eq. 3:

$$f_\alpha = \sum_i^m \mathcal{F}_\alpha(f_I, f_{A_i}) P_{O_i}, \qquad (4)$$

where $P_{O_i}$ is the *prior* category probability ($P(O_i)$ in Eq. 3). Eq. 4 indicates the attribute *instantiation* from $A$ to $\alpha$ with $I$ as the *condition*. Hence, we can equally translate the $\alpha$ estimation problem into a **conditioned instantiation problem**. $\mathcal{F}_\alpha(\cdot)$ is implemented with multi-head attention mechanism (Vaswani et al., 2017), as shown in Fig. 5. Both $f_{A_i}$ and $f_I$ are fed to three independent linear transformations to obtain key, query, and value vectors $K, Q, V$ and $K', Q', V'$. The attention output is $softmax([Q, Q']^T[K, K']/\sqrt{d})[V, V']^T$ where $d$ is the feature dimension. The output is compressed by a linear layer to the instantiated representation $\mathcal{F}_\alpha(f_I, f_{A_i})$. The debiased representation $f_\alpha$ is the expectation of $\mathcal{F}_\alpha(f_I, f_{A_i})$ w.r.t $P_{O_i}$ according to back-door adjustment (Eq. 3).

### 4.2.2 AFFORDANCE ESTIMATION

Similar to $\alpha$, in $\beta$ estimation, category bias also exists:

$$P(\beta|I, \alpha) = \sum_i^m P(\beta|I, \alpha, O_i) P(O_i|I, \alpha). \qquad (5)$$

With Eq. 3, $\alpha$ can be seen as "enforced" and deconfounded, as it is beforehand estimated. For $I$, we again use intervention (Pearl et al., 2016):

$$P(\beta|do(I, \alpha)) = \sum_i^m P(\beta|I, \alpha, B_i) P(O_i). \qquad (6)$$

**Affordance Instantiation.** Similar to Eq. 3, $P(B_j|O_i) = 1$ iif $i = j$, $P(B_j|O_i) = 0$ if $i \neq j$, we omit the process for simplicity. An FC is used to obtain $f_B$ and Eq. 6 is implemented as:

$$f_{B_i} = \mathcal{F}_B(\bar{f}_{O_i}, f_{A_i}), f_\beta = \sum_i^m \mathcal{F}_\beta(f_I, f'_\alpha, f_{B_i}) P_{O_i}. \tag{7}$$

As shown in Fig. 5, for causal inference operation (Sec. 5), $f_\alpha$ is first separated to attribute category-wise features $f_{\alpha_p}$ by multiple independent FCs, where $f_{\alpha_p}$ is the feature of $p^{th}$ attribute. The features are aggregated via concatenating-compressing by an FC to $f'_\alpha$. In this way, we can manipulate specific attributes of $f'_\alpha$ by masking some certain $f_{\alpha_p}$, in order to evaluate the causal learning of our model. We compare different aggregation methods in the ablations. And $f_B = \{f_{B_i}\}_{i=1}^m$. Given the conditions $\{f_I, f'_\alpha, f_{B_i}\}$, $\mathcal{F}_\beta(\cdot)$ operate the instantiation. $\mathcal{F}_B(\cdot), \mathcal{F}_\beta(\cdot)$ are implemented the same as $\mathcal{F}_A(\cdot), \mathcal{F}_\alpha(\cdot)$, except the first input condition of $\mathcal{F}_\beta$ is the concatenation of $f'_\alpha, f_I$.

**Learning Objectives.** To drive the learning, we devise several objectives.

**(1) Category-level loss $L_C$.** After obtaining the category-level representations $f_A, f_B$, we input them to two linear-Sigmoid classifiers $g_A, g_B$ to classify $A, B$, given GT $O$ in training: $\mathcal{P}_{A_i} = g_A(f_{A_i}), \mathcal{P}_{B_i} = g_B(f_{B_i})$. The binary cross-entropy losses of the $i$-th object category are represented as $L_{A_i}$ and $L_{B_i}$. So the total category-level loss is $L_C = \sum_{i=1}^m (L_{A_i} + L_{B_i})$.

**(2) Instance-level loss $L_I$.** As for the instance-level, we send $f_\alpha$, $f_\beta$, together with the features before expectation $\mathcal{F}_\alpha(f_I, f_{A_i}), \mathcal{F}_\beta(f_I, f'_\alpha, f_{B_i})$ to linear classifiers $g_\alpha$ and $g_\beta$. The binary cross-entropy losses of these output logits are represented as $L_\alpha, L_\beta$. The separated attribute features $f_{\alpha_p}$ are also sent to independent binary classifiers $g_{\alpha_p}$ and computed losses with ground truth of $p^{th}$ attribute, which is included in $L_\alpha$ The instance-level loss is then $L_I = L_\alpha + L_\beta$.

Finally, the total loss is $L = \lambda_C L_C + L_I$. We adopt a two-stage policy: first inferring attributes, then reasoning affordances.

## 5 EXPERIMENT

**Metric**. For $\alpha, \beta$ learning, as an object can have multiple attributes and affordances (multi-label classification), we use *mean Average Precision (mAP)*. For causal reasoning, we adopt TDE (Pearl et al., 2016) for evaluation. The general idea of TDE is to assess the effect of a variable: constructing the **counterfactual** scenario via eliminating the effect of the controlled variable, then subtracting the counterfactual from the original. Give an intuitive example, the effect of cargo can be measured by the speed difference between a loaded car and an empty car. In OCL, we formulate TDE as the **affordance probability change** $\Delta P_\beta$. Here, we redefine $\Delta P_\beta$ as the difference between $P(\beta_q)$ and $P_{TDE}^{\alpha_p}(\beta_q)$: ($p \in [1, 114], q \in [1, 170]$): comparing the original with its counterfactual without the effect of attribute $\alpha_p$. For an object, let $P(\beta_q)$ denote its original probability of $\beta_q$ estimated by a trained model. As we use learned *attribute feature* to infer $\beta_q$, we assign zero-mask (Tang et al., 2020b) to the feature of $\alpha_p$ and keep the other attribute features, to get $P_{TDE}^{\alpha_p}(\beta_q)$ with the *same* model parameters. After concurrently operating TDE for all $[\alpha_p, \beta_q]$, we obtain a huge cube of *instance-attribute-affordance*, where each grid indicates the assessed *effect* of certain $[\alpha_p, \beta_q]$ for *each instance*. For each $[\alpha_p, \beta_q]$, We score instances following (*Appendix Sec. A.4*):

**(1) TDE**: If $[\alpha_p, \beta_q]$ has annotated causality, when eliminating the effect of $\alpha_p$, $|P(\beta_q) - P_{TDE}^{\alpha_p}(\beta_q)|$ should be large. If $[\alpha_p, \beta_q]$ does not have causality, the difference should be small. Beside the change scale, the right change direction is also expected. We set TDE score $TDE_{\beta_q}^{\alpha_p}$ as:

$$TDE_{\beta_q}^{\alpha_p} = \begin{cases} max(P(\beta_q) - P_{TDE}^{\alpha_p}(\beta_q), 0), & GT_{\beta_q} = 1, \\ max(P_{TDE}^{\alpha_p}(\beta_q) - P(\beta_q), 0), & GT_{\beta_q} = 0, \end{cases} \tag{8}$$

where $GT_{\beta_q}$ is the label of $\beta_q$.

**(2) $\alpha$-$\beta$-TDE**: $\alpha_p, \beta_q$ should alse be infered accurately. We multiply $TDE_{\beta_q}^{\alpha_p}$ with $P(\alpha_p = GT_{\alpha_p}) * P(\beta_q = GT_{\beta_q})$ as a unified score. We compute AP for each $[\alpha_p, \beta_q]$ and average them to mAP.

**Baselines**. We list all baselines here briefly (detailed in *Appendix Sec. A.5*): (1) Direct Mapping from $f_I$ to $P_\alpha, P_\beta$ (DM-V); (2) DM from Attribute (DM-At); (3) DM from Attribute and Object

| Method | $\alpha$ | $\beta$ | TDE | $\alpha$-$\beta$-TDE |
|---|---|---|---|---|
| DM-V | 29.9 | 51.8 | - | - |
| DM-L | 21.2 | 47.5 | - | - |
| MM | 23.8 | 48.9 | - | - |
| LingCorr | 7.9 | 25.9 | - | - |
| KPMF | 25.4 | 49.1 | - | - |
| $A$&$B$-Lookup | 18.9 | 30.9 | - | - |
| HMa | 28.6 | 51.7 | - | - |
| DM-At | 27.9 | 52.3 | 7.6 | 6.7 |
| DM-AtO | 28.0 | 52.2 | 8.1 | 7.0 |
| Ngram | 22.6 | 50.8 | 8.3 | 7.6 |
| MLN-GT | - | 33.4 | **9.5** | 9.1 |
| Attention | 24.1 | 48.7 | 8.1 | 7.1 |
| OCRN | **31.6** | **53.3** | 9.5 | **9.2** |
| DM-At w/ $L_{TDE}$ | 28.4 | 52.2 | 15.5 | 14.0 |
| DM-AtO w/ $L_{TDE}$ | 28.0 | 52.4 | 15.4 | 13.6 |
| Ngram w/ $L_{TDE}$ | 22.2 | 49.9 | 14.1 | 12.9 |
| MLN-GT w/ $L_{TDE}$ | - | 33.7 | 12.3 | 11.8 |
| Attention w/ $L_{TDE}$ | 23.9 | 49.0 | 17.8 | 15.5 |
| OCRN w/ $L_{TDE}$ | **31.5** | **53.6** | **20.3** | **16.9** |

Table 2: OCL results. w/ $L_{TDE}$ indicates that training with TDE loss. The baselines in the upper block cannot operate TDE due to the model structure (*Appendix Sec. A.5*).

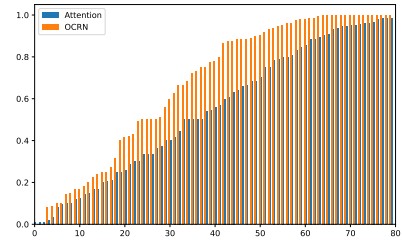

(a) Proportion of correct TDE when $GT_{\beta_q} = 0$.

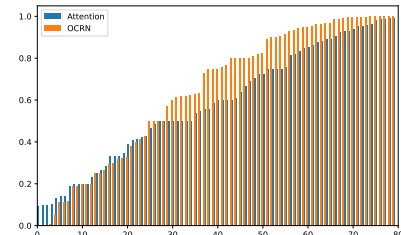

(b) Proportion of correct TDE when $GT_{\beta_q} = 1$.
Figure 6: TDE performance of different $[\alpha_p, \beta_q]$.

(DM-AtO); (4) DM from Bert (Devlin et al., 2018) Linguistic Representation (DM-L); (5) Visual-Linguistic Representation, i.e., Multi-Modality (MM); (6) Markov Logic Network (Richardson & Domingos, 2006), using **GT** $\alpha$ to infer $\beta$ (MLN-GT); (7) Linguistic Correlation of $O$-$\alpha$, $O$-$\beta$ (LingCorr); (8) Kernelized Probabilistic Matrix Factorization (KPMF) (Zhou et al., 2012); (9) **A**&**B** Lookup, directly getting $P_A, P_B$ from $M'_A, M'_B$ as $P_\alpha, P_\beta$; (10) Cause as Attention (Attention); (11) Hierarchical Mapping from $I$ to $A/B$ then to $\alpha/\beta$ (HMa). (12) Ngram (Lin et al., 2012).

**TDE loss**. Though the machine is expected to learn the causalities given $\alpha, \beta$ labels only. We also wonder how it would perform given causal supervision. We adopt an extra Hinge loss to maximize the TDE score of all $[\alpha_p, \beta_q]$. In detail, without the effect of one certain $\alpha_p$, we intend the probability of causal-related $\beta_q$ to keep farther from $GT_{\beta_q}$ by a margin $\tau$ (= 0.1 in experiments), formulated as:

$$TDE_{\beta_q}^{\alpha_p} = \begin{cases} \max\{0, \tau - (P(\beta_q) - P_{TDE}^{\alpha_p}(\beta_q))\}, & GT_{\beta_q} = 1, \\ \max\{0, \tau - (P_{TDE}^{\alpha_p}(\beta_q) - P(\beta_q))\}, & GT_{\beta_q} = 0. \end{cases} \quad (9)$$

We enumerate all annotated $[\alpha_p, \beta_q]$ of an instance to obtain $L_{TDE}$. Different from the default, the total loss here is $L = \lambda_C L_C + L_I + \lambda_{TDE} L_{TDE}$.

**Implementation Details**. For a fair comparison, all methods adopt a shared COCO (Lin et al., 2014) pre-trained ResNet-50 (He et al., 2016) (frozen) to extract $f_I$ and use the same object boxes in training and inference. In OCRN, the dimension of $f_I$ and all representations $f_{A_i}, f_{B_i}, f_\alpha, f_\beta$ is 1024. The individual features of each attribute category is 512d and aggregated to 1024d by an FC. We train the attribute module with a learning rate of 0.3 and batch size of 1024 for 470 epochs. Then the attribute module is frozen, and the affordance module is trained with a learning rate of 3.0e-3 and batch size of 768 for 20 epochs. In training, $\lambda_C = 0.03$, $\lambda_{TDE} = 3$.

**Results**. Tab. 2 presents the results. OCRN outperforms the baselines and achieves decent improvements on all tracks. In terms of $\alpha$, OCRN largely outperforms DM-V with 1.7 mAP (default) and DM-At with 3.1 mAP (w/$L_{TDE}$). As for $\beta$, OCRN also achieves 1.0 mAP (default) and 1.4 mAP (w/$L_{TDE}$) improvements compared to DM-At and DM-AtO w/$L_{TDE}$. Comparatively, HMa utilizes the supervision of $A, B$, but it performs much worse. $A$&$B$ Lookup directly uses GT $A, B$ to infer $\alpha, \beta$, but its poor performance verifies the significant difference between $A, B$ and $\alpha, \beta$. Moreover, we find that all methods perform better on $\beta$ than $\alpha$. This may because $\alpha$ are more diverse than $\beta$, e.g., we can `eat` lots of `foods`, but `foods` usually have various attributes (`fruit` v.s. `pizza`). Another reason is that the positive samples in $\beta$ labels (23.2%) are much more than the positives in $\alpha$ labels (9.4%). The different pos-neg ratio affect the learning a lot and result in the above gap.

In TDE evaluation, without the guidance of $L_{TDE}$, all methods achieve unsatisfied performances. However, OCRN still has an advantage. Only MLN adopting the first-order logic and GT $\alpha$ labels is

| Method | $\alpha$ | $\beta$ | TDE | $\alpha$-$\beta$-TDE |
|---|---|---|---|---|
| OCRN | **32.4** | **52.2** | **20.5** | **17.0** |
| w/o deconfounding | 32.1 | 51.8 | 18.2 | 16.1 |
| w/o $L_{A_i}, L_{B_i}$ | 32.1 | 51.8 | 19.8 | 16.7 |
| w/o $L_\alpha, L_\beta$ | 10.0 | 27.0 | 16.6 | 16.4 |
| 128 Dims | 31.7 | 51.5 | 18.0 | 16.0 |
| 512 Dims | 32.3 | 52.1 | 19.9 | 16.7 |
| 2048 Dims | 32.2 | 51.5 | 19.1 | 16.3 |
| Mean aggregation | 32.2 | 51.3 | 18.9 | 16.7 |
| Max-pooling aggregation | 32.1 | 49.1 | 19.0 | 16.8 |
| Random counterfactual | 32.4 | 51.8 | 5.1 | 5.1 |

Table 3: Ablation study results on the val set.

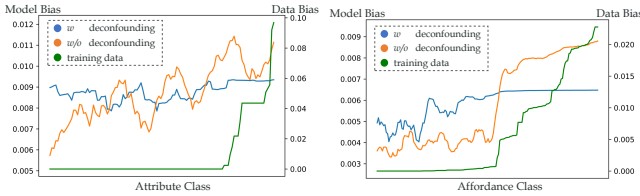

Figure 7: Attribute bias (w/ and w/o deconfounding) for category `frying pan`.

Figure 8: Affordance bias (w/ and w/o deconfounding) for category `giraffe`.

comparable with OCRN. If trained with $L_{TDE}$ and direct causality labels, all methods perform much better to learn complex causalities. Particularly, the typical deep learning model Attention performs best in baselines, but MLN no longer holds the previous advantage. Relatively, OCRN shows greater improvements and outperforms Attention with 2.5 and 1.4 mAP on two TDE tracks. Due to the page limit, we provide more analyses in *Appendix Sec. A.6*. In *Appendix Sec. A.7*, we further apply OCRN to Human-Object Interaction Detection (Chao et al., 2018), where OCRN boosts the performance and verifies the efficacy and generalization of OCL.

**Visualization. (1)** In Fig. 6, we show the correct instance proportions (%) of OCRN and Attention after TDE. (a) depicts the randomly chosen causal $[\alpha_p, \beta_q]$ pairs with $GT_{\beta_q} = 1$ expecting $P(\beta_q) > P^{\alpha_p}_{TDE}(\beta_q)$. (b) depicts the randomly chosen causal $[\alpha_p, \beta_q]$ pairs with $GT_{\beta_q} = 0$ expecting $P(\beta_q) < P^{\alpha_p}_{TDE}(\beta_q)$. The higher proportions indicate that OCRN performs better than Attention on TDE. **(2)** To verify the deconfounding, we compare the model bias of OCRN **w/ or w/o deconfounding**. The bias of category $O$ upon an attribute $\alpha$ is measured following Zhao et al. (2017), by $b(O, \alpha) = c(O, \alpha) / \sum_{\alpha'} c(O, \alpha')$. When measuring **data** bias, $c(O, \alpha)$ is the number of co-occurrence of $O$ and $\alpha$ in OCL, and when it comes to **model** bias, $c(O, \alpha)$ is the sum of probabilities that $O$ are predicted positive with $\alpha$. The bias of $\beta$ is measured in the same manner. Fig. 7 and 8 show some examples of the biases of training data and models, indicating that OCRN deconfounding effectively prevents the model from bias toward the train set.

**Ablation Study**. We conduct ablation studies to verify the components of OCRN w/$L_{TDE}$ (Tab. 3) on the val set. **(1) Deconfounding.** OCRN w/o deconfounding is implemented following Eq. 2 and 5, where $P(O|I)$ and $P(O|I, \alpha)$ are the category predictions of pre-trained detectors (Liu et al., 2021). Both $\alpha$ and $\beta$ performances significantly decline, as explained by Fig. 7 and 8. **(2) Losses.** The performances slightly drop after removing category-level $L_{A_i}, L_{B_i}$, but significantly drop without instance-level $L_\alpha, L_\beta$. **(3) Feature dimension.** For $f_{A_i}, f_{B_i}, f_\alpha, f_\beta$, smaller and larger feature sizes all have degrading effects. **(4) TDE-related implementations.** We probe some different methods: (a) Mean aggregation: $f'_\alpha = \sum_i f_{\alpha_p}$; (b) Max-pooling aggregation: $f'_\alpha$ is the max value of $f_{\alpha_p}$ as each component; (c) Random counterfactual feature: assigned random vector as the counterfactual attribute feature (instead of zero vector) during TDE. These methods perform worse than the chosen setting on TDE performance but are comparable on $\alpha$ and $\beta$ performance.

**Discussion**. Overall, OCL poses extreme challenges to current AI systems. It expects representative learning to accurately recognize attribute and affordance from raw data meanwhile causal inference to capture the causalities within diverse instances and contexts, i.e., both the *intuitive System 1 and logical System 2* (Bengio, 2019). From the experiments, we find that models struggle to achieve satisfying results on all tracks simultaneously. Notably, it is difficult to achieve a satisfying TDE score via data fitting, and there is much room for improvement. For future studies, a harmonious performance on $\alpha, \beta$, and causality learning are encouraged to capture object knowledge better. Potential directions may include *causal representation learning* (Schölkopf et al., 2021), *casual reinforcement learning* (Bareinboim, 2020), *neural-symbolic reasoning* (Besold et al., 2017), etc.

# 6 CONCLUSION

In this work, we introduce object concept learning (OCL), which expects machines to infer affordances and explain what attributes enable an object to possess these affordances. Accordingly, we build an extensive real-world dataset and present a baseline OCRN based on casual intervention and instantiation. OCRN achieves decent performance and follows the causalities well. However, OCL remains challenging and would inspire a line of studies upon reasoning-based object understanding.

## 7 ETHICS STATEMENT

This work primarily proposes a daily object dataset to advance machine understanding of objects for the future study of embodied AI, intelligent robots, etc. Our work may help the home service robots better understand the human living scenes and then help to improve our lives, especially for the health care of patients, elders, and children. All the data used in OCL are publicly available. We have cited the creators and followed the official licenses. For all images, we only provide the URL links in our future web for research uses only. Moreover, modern deep learning models are usually computationally expensive. Our proposed model OCRN also needs multiple GPUs to train. In the future, OCL encourages a new line of studies to understand objects' common sense and combine causal reasoning and deep learning. We believe the future works focusing on learning the causality instead of heavily fitting data would be more computationally lite, less data-hungry, and low-carbon.

## 8 REPRODUCIBILITY STATEMENT

We provide our source code, OCL dataset samples, and a video demo in the supplementary material (the zip file). It includes all the manuscripts to preprocess and load data, construct the OCRN model with the hyper-parameters, train and test the model, evaluate the performance on the OCL benchmark, etc. Please refer to the code file for more details. The details of related datasets, OCL data processing and annotation are in the main text Sec. 3, Appendix Sec. A.3, A.11. Our code and data will be publicly available.

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

# A APPENDIX

We report more details and analyses about OCL in the appendix:

## A.1 CATEGORY/ATTRIBUTE/AFFORDANCE SELECTION

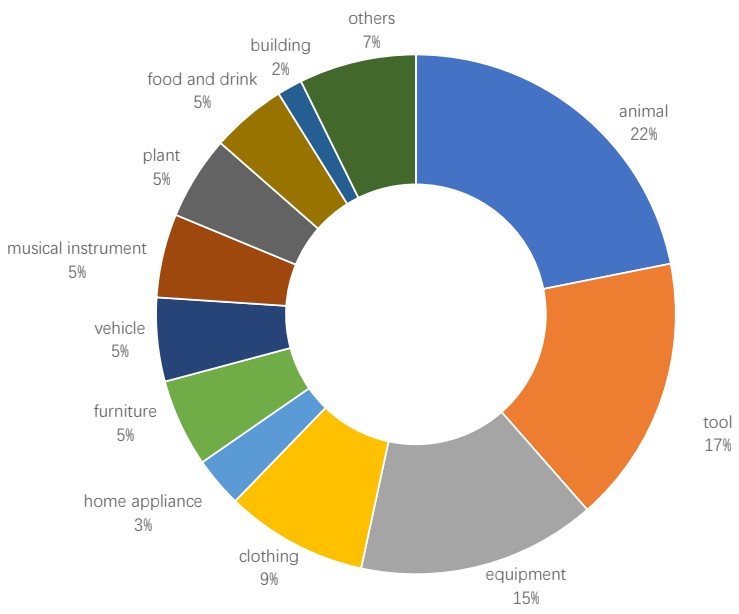

Figure 9: Super-categories of objects in OCL.

We first choose affordances, categories, and attributes, considering their causal relations. We present their word cloud in Fig. 10. The complete lists can be found in Appendix Sec. A.11.

(1) **Affordance**: To build a general and applicable knowledge base, we collect 1,006 affordance candidates from several widely-used action or affordance datasets (Chao et al., 2015; Gu et al., 2018; Chao et al., 2018; Gupta & Malik, 2015; Zhu et al., 2014; Nguyen et al., 2017) (957 from Chao et al. (2015), 160 from Gu et al. (2018), 146 from Chao et al. (2018), 97 from Gupta & Malik (2015), 41 from Zhu et al. (2014), 21 from Nguyen et al. (2017), with *overlaps*). We find that not all affordances are in common use and some of them are difficult for visual recognition, e.g., accept (consider right and proper). So each candidate is scored by 5 human experts from 0.0 to 5.0 according to generality and commonness. We keep **170** top-scored affordances in our base (134

(a) Category        (b) Attribute        (c) Affordance

Figure 10: Word clouds of categories, attributes, and affordance (by positive frequencies in OCL).

from Chao et al. (2015), 78 from Gu et al. (2018), 127 from Chao et al. (2018), 53 from Gupta & Malik (2015), 13 from Zhu et al. (2014), 11 from Nguyen et al. (2017), with *overlaps*).

(2) **Category**: Considering the taxonomy (WordNet (Fellbaum, 2012)), we collect a pool with over 1,742 object categories from previous datasets (Farhadi et al., 2009; Xiao et al., 2010; Patterson & Hays, 2016; Liu et al., 2017) (32 from Farhadi et al. (2009), 28 from Patterson & Hays (2016), 717 from Xiao et al. (2010), 1000 from Liu et al. (2017), with *overlaps*). Then we merge the similar categories according to WordNet (Fellbaum, 2012), and filter out the categories which are not common daily objects (e.g. "man", "planet"), unrelated to the above 170 affordances (e.g. "skyscraper") or too uncommon (e.g. "malleefowl"). Finally our database has **381** common object categories. These object categories are divided into **12** super categories, showed in Fig. 9.

(3) **Attribute**: We extract the attribute classes from several large-scale attribute datasets (Farhadi et al., 2009; Xiao et al., 2010; Patterson & Hays, 2016; Liu et al., 2017; Krishna et al., 2016) (64 from Farhadi et al. (2009), 203 from Patterson & Hays (2016), 66 from Xiao et al. (2010), 25 from Liu et al. (2017), top 500 from Krishna et al. (2016)), and manually filter the 500 most frequent attributes. Five experts give 0.0 to 5.0 scores based on their relevance to human actions and the selected 170 affordances to better explore the causal relations between attributes and affordances. Some attributes (e.g. `cloudy`, `competitive`) that are not helpful for affordance reasoning are discarded. Finally, **114** attributes are kept, covering colors, deformations, supercategories, various surface, geometrical, and physical properties.

## A.2    ANNOTATION DETAILS

### Attribute Annotation.

(1) **Category-level attribute** ($A$). Following Osherson et al. (1991), to avoid bias, annotators are given *category-attribute pairs* (category *names*, not images). They propose a 0-3 score according to the category concept in their minds (0: No, 1: Normally No, 2: Normally Yes, 3: Yes). Each pair is annotated by three annotators and takes the plurality as $A$ label. If the range of 3 proposals exceeds 1, another three annotators will re-annotate this pair until achieving consensus. We binarize the annotations (0: No, 1: Yes) with a threshold of 2 and get a category-level attribute matrix $M_A$ ($[381, 114]$).

(2) **Instance-level attribute** ($\alpha$). Two annotators label each pair with 0 (No) and 1 (Yes). If they give different labels, this pair will be handed over to another two annotators until meeting consensus.

### Affordance Annotation.

(1) **Category-level affordance** $B$. Following Chao et al. (2015), the annotators are given category-affordance pairs. The pairs are annotated in four bins (0-3) and normalized (same as $A$) to describe the possibility of an affordance in a category. Each pair is annotated by three annotators and makes consensus the same as $A$. The 0-3 scores are binarized (1: Yes, 0: No) with a threshold of 2. The final category-level affordance matrix $M_B$ is $[381, 170]$.

(2) **Instance-level affordance** $\beta$ is annotated for **every instance** with the help of *object states* (Isola et al., 2015). As $B$ is determined by common states, objects in specific states may have different affordances from $B$, e.g., we cannot `board` a `flying` plane. Moreover, if a service robot finds a `broken cup`, it may infer that the `cup` can still hold water as it is trained with $B$ labels. Thus, we need detailed $\beta$ beyond $B$. As the instances in the same state should have similar $\beta$ (all `rotten apples` cannot be `eaten`), six experts first conclude the states. The experts scan all instances of each category and use their knowledge of affordance to define all the existing states. Then all 186 K instances are dispatched to the concluded states via crowdsourcing. If some instances do not belong

to any predefined states, they will be returned to the experts to add more states. In total, **1,376** states are defined, and each category has 3.6 states on average. Next, $\beta$ is annotated for each state. Given a *state-affordance pair* and example images, two annotators mark it with 0 (No) and 1 (Yes). The results are combined in the same way as $\alpha$. Thus, each instance would have a state and the corresponding $\beta$. An annotator would recheck each instance together with its state and $\beta$ labels to ensure the quality. If its state is inaccurate or the state $\beta$ labels are unsuitable, this annotator would correct them.

**Causal Relation Annotation.**

**(1) Filtering**. Starting from the [114,170] matrix of $\alpha$-$\beta$ classes, we ask three experts to vote the causal relation of each class. They scan all instances to answer whether the relation exists in any cases. Finally, we obtain about 10% $\alpha$-$\beta$ classes as candidates.

**(2) Instance-level causality**: we also adopt object states as a reference. For each *state-$\alpha$-$\beta$* triplet, two annotators are asked whether the specific attribute is the *direct* cause of this affordance in this state and gives their binary answer. We use the same method in annotating $\beta$ to combine results and assign state-level labels to instances. Next, for all instances of a state, an expert decides whether the state-level relations are reasonable for each instance in specific contexts and correct the inaccurate ones. Finally, we obtain about 2 M *instance-$\alpha$-$\beta$* triplets of causal relations.

**A Running Example of Dataset Construction.**

A running example is shown in Fig. 11 to show the process of annotations clearly.

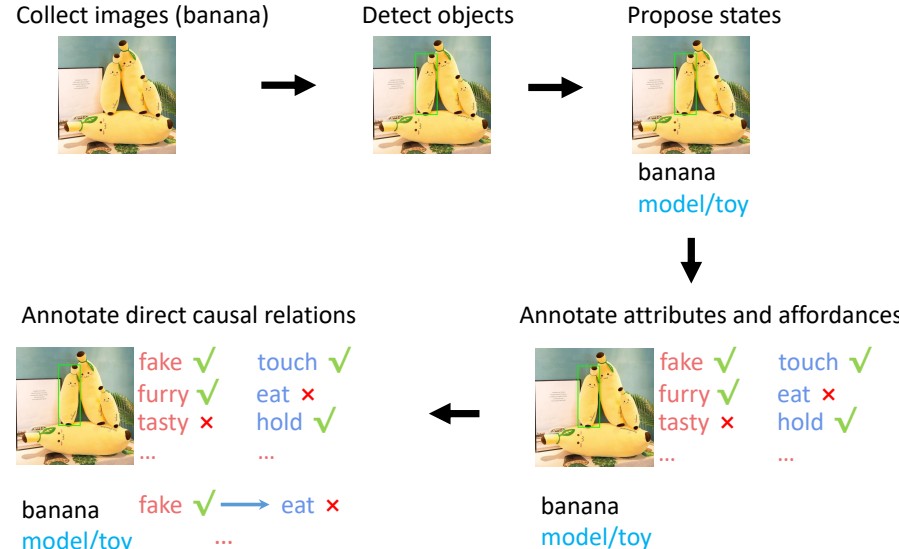

Figure 11: A running example of dataset construction.

## A.3 OCL Characteristics

### A.3.1 Object Box Size

We visualize the distribution of normalized object box size in Fig. 12, where the box width and height are normalized by the width and height of the whole image. It shows that most objects in our knowledge base are *small objects*, providing abundant regional information.

### A.3.2 Annotator Information

Annotators' age, major, and education degree are presented in Fig. 13, 14, and 15.

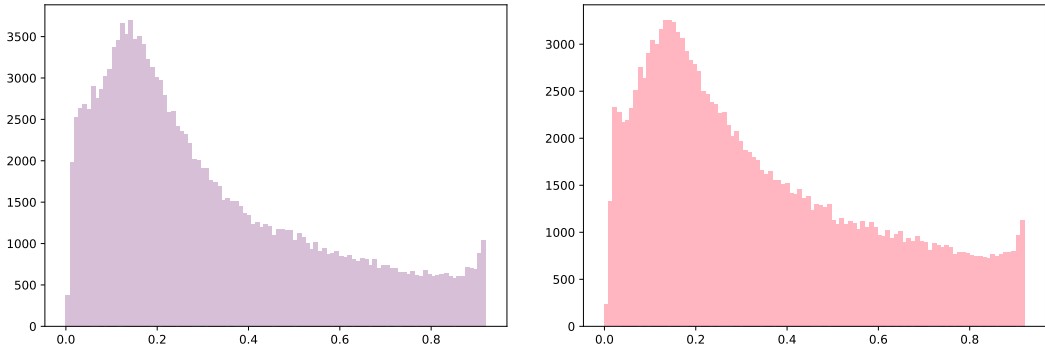

Figure 12: Distribution of normalized object box width (left) and height (right).

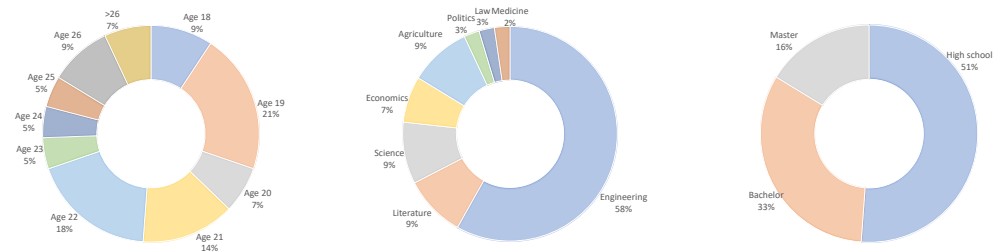

Figure 13: Age information of annotators.

Figure 14: Major information of annotators.

Figure 15: Degree information of annotators.

### A.3.3 MATRIX SAMPLES

The category-level attribute and affordance $(A, B)$ matrices are detailed in Fig. 16, 17 as heatmaps, the cells with dark color indicate positive samples. For example, `ice cream` is `cold` while `clock` is not `natural`, `cake` can be `eaten` while `eraser` can not be `cooked`. These are in line with our common sense.

### A.3.4 STATE DISTRIBUTION

Before annotating the affordances, we first define the object states for all object categories and annotate the state affordances. In total, we define 1,376 states for 381 object categories. The state list can be found in Appendix Sec. A.11.3. And Fig. 18 shows the state distribution per object category.

### A.3.5 ATTRIBUTE-AFFORDANCE RELATION

We analyze the instance-level attribute-affordance relations in our knowledge base under three criteria. (1) **Attribute Conditioned Affordance Probability.** It's computed as $P(\beta|\alpha)$ to estimate affordance probability given an attribute. The range is [0,1]. (2) **Attribute-Affordance Correlation.** For all instances in our dataset, we evaluate the label correlation of each attribute-affordance pair, whose scale is in [-1,1]. (3) **Attribute-Affordance Causality**. Start with the annotated cause-effect $(\alpha - \beta)$ labels, we count for how many times each attribute-affordance pair appear in our dataset and normalize the value by the maximum occurrences, leading to value in the range [0,1]. It should be mentioned that we only annotate whether an attribute-affordance pair **has** explicit and key causality, but the detailed effect (positive or negative) should be referred to instance labels.

We visualize the samples of attribute-affordance relation matrices in Fig. 19, 20, 21 and observe some interesting properties of them. They reveal some common relations, such as what between *tasty* and *eat*. However, some of criteria suffer from data bias. For the condition matrix in Fig. 19, it only cares about cases with positive attribute labels, which is not good in highlighting the negative

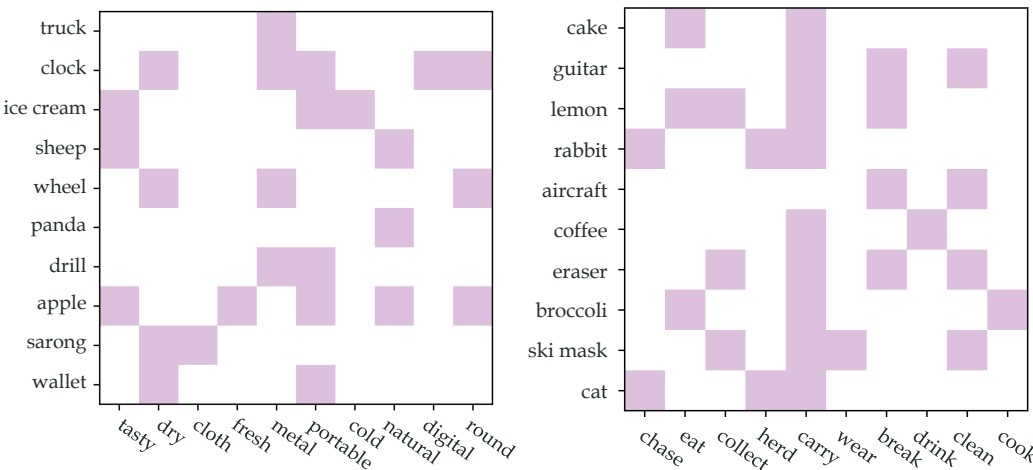

Figure 16: Category-level attribute ($A$) matrix.  Figure 17: Category-level affordance ($B$) matrix.

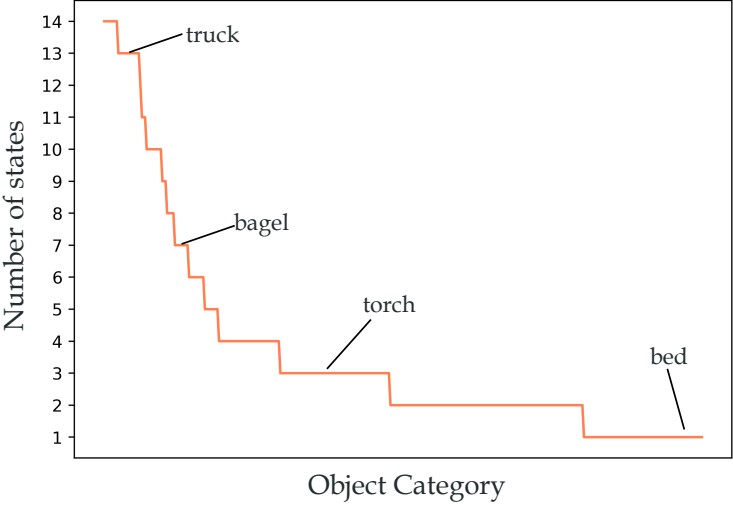

Figure 18: State distributions of different object categories.

relations, e.g. the relation between *natural* and *produce*. For the former two matrices in Fig. 19, 20, they all point out the relation between *tasty* and *pick*, since most *tasty* objects are *pickable food*. This finding is simply misled by the data bias but violate the causal graph (inferece from attribute to object category, then affordance). Last, the matrix obtained from our causal annotation in Fig. 21 is more sparse, clear of causality.

### A.3.6 UNIFIED OBJECT REPRESENTATION

To compare attribute-only and attribute-affordance representation abilities, we cluster the object instances of two similar animals (zebra and horse) with their attribute labels and attribute-affordance labels, respectively. The results are shown in Fig. 22 via t-SNE (Maaten & Hinton, 2008). With both attribute and affordance labels, zebra and horse can be better separated than attribute only. And attribute and affordance together can differentiate specific **states** well, such as `riding`, `pulling car`, etc.

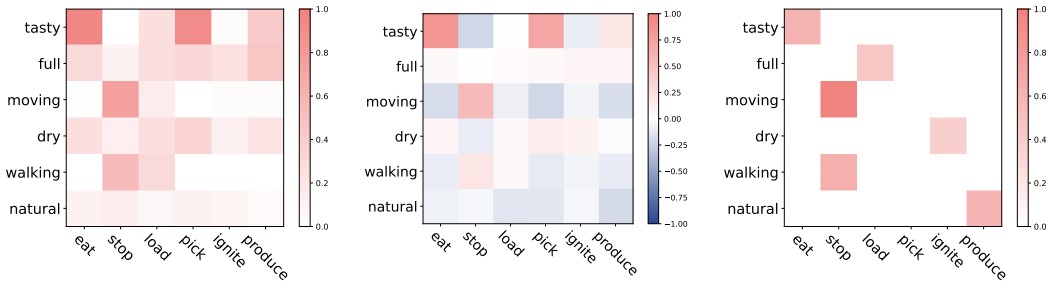

Figure 19: Attribute conditioned affordance matrix.

Figure 20: Attribute-affordance **correlation** matrix.

Figure 21: Attribute-affordance **causality** matrix.

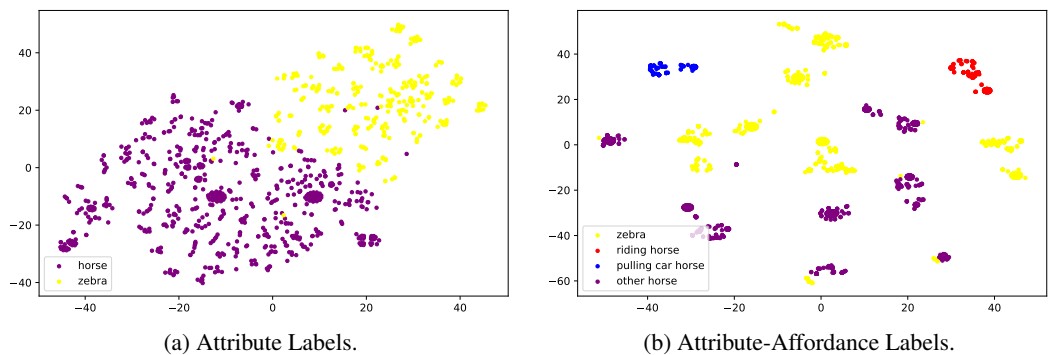

(a) Attribute Labels.

(b) Attribute-Affordance Labels.

Figure 22: Clustering using attribute and attribute-affordance labels.

### A.3.7 DIFFERENCE BETWEEN CATEGORY- AND INSTANCE-LEVEL LABELS

We analyze the differences between category-level $A, B$ labels and instance-level $\alpha, \beta$ labels. For each object category, we compute the *average ratio* of changed attribute/affordance classes during each instantiation from $A$ to $\alpha$ or from $B$ to $\beta$. The top-50 categories with the most significant differences between $A$ and $\alpha$ as well as $B$ and $\beta$ are reported respectively in Fig. 23. We find that affordance labels change more dramatically than attribute labels during instantiations. This is because **each** attribute change may affect **several** affordances, e.g., when a common book becomes wet, we can neither open nor read it.

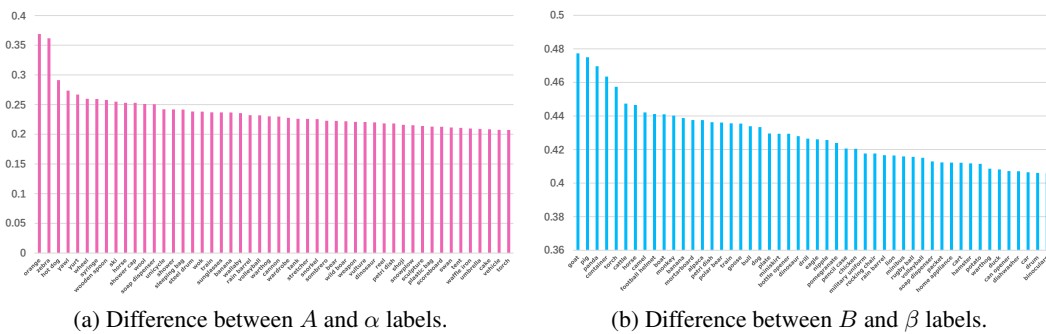

(a) Difference between $A$ and $\alpha$ labels.

(b) Difference between $B$ and $\beta$ labels.

Figure 23: Top-50 object categories with largest ratio of difference between category- and instance-level labels.

### A.3.8 ATTRIBUTE-AFFORDANCE CAUSAL RELATIONS

We annotate all object instances' causal relations of filtered $[\alpha_p, \beta_q]$ pairs. In total, 1,085 $[\alpha_p, \beta_q]$ pairs are chosen for the causality annotation, and over 2 M *instance-$\alpha$-$\beta$* triplets are annotated. In TDE evaluation (main text Sec. 5), we report the mean AP of top-300 $[\alpha_p, \beta_q]$ pairs to avoid the biased influence of very rare $[\alpha_p, \beta_q]$ pairs that include less than 35 object instances.

### A.3.9 DATA PARTITIONING

For the OCL task, our knowledge base is split into train, val, and test sets. The statistical details of the split are listed in Tab. 4. The image number ratio of the three sets is nearly 4:1:0.6, and the instance ratio is around 5:1:1.

| Set | Image | Object Instance | Object category |
|---|---|---|---|
| Train | 56,916 | 135,148 | 381 |
| Val | 14,446 | 25,176 | 221 |
| Test | 9,101 | 25,617 | 221 |
| Val+Test | 23,547 | 50,793 | 221 |
| All | 80,463 | 185,941 | 381 |

Table 4: Detailed data split of our knowledge base.

### A.3.10 IMAGES AND INSTANCES

Some additional data samples of our knowledge base are shown in Fig. 24, 25a, 25b, 26, and 27, including samples of diverse object categories with various bounding box distributions, different attributes and affordances, and human-labeled object states and obvious causal relations.

We also detail the image and object instance counts in Fig. 28, 29, and 30.

### A.3.11 MORE STATISTICS OF ANNOTATION

We divide $A, B, \alpha, \beta$, causality annotation into multiple finer-grained small sets in our pipeline. Generally, we have 13, 19, 124, 140, 85 annotators-sets (381 totally) for $A, B, \alpha, \beta$, causality annotation respectively. We assign each small set to 2 annotators. However, considering the controversial situations introduced, part of the annotation are confused cases based on their results. In the whole process, 9.6% of $A$, 7.7% of $B$, 5.2% of $\alpha$, 7.9% of $\beta$, and 13.7% of causality are confusing and re-assigned to additional annotators. These indeterminable ones will be sent to two extra two annotators until making an agreement. The quality of the dataset is guaranteed by a low confusion ratio and multiple refining stages.

### A.3.12 POTENTIAL BIAS

We have considered the bias issue in the construction of our dataset. 1. In our dataset, the existing datasets (ImageNet Deng et al. (2009), COCO Lin et al. (2014), aPY Farhadi et al. (2009), SUN Xiao et al. (2010)) are open-sourced datasets and the images collected from Internet are publicly accessible too. The dataset is constructed for only non-commercial purposes. We will only provide URLs of these images to avoid copyright infringement. 2. During image collection, we choose images with general objects and are particularly careful with the images selection to avoid unsuitable content, private images, or implicit biases. 3. During annotation, the annotators cover different genders, ages, and fields of expertise to avoid potential annotation biases. And they are all informed how we will use the annotations in our research.

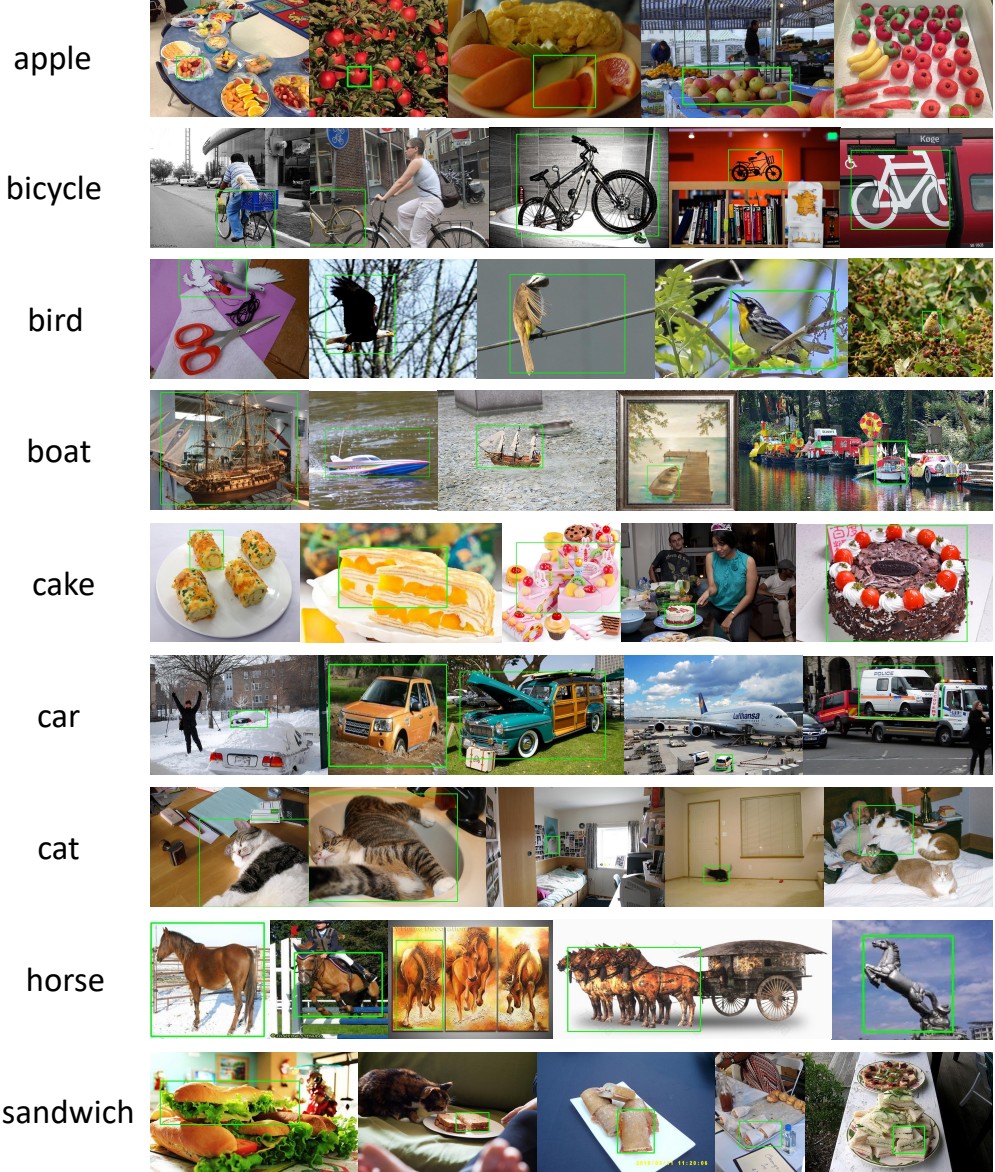

Figure 24: More OCL samples of objects.

## A.4 TDE METRIC DETAILS

TDE is to measure whether a model infers affordance with proper attention to the causality-related attribute. That said, when removing the attribute, the model is expected to have *large prediction difference further away from the ground truth*. We detail some settings in our TDE metric here. For the TDE score:

$$TDE_{\beta_q}^{\alpha_p} = \begin{cases} max(P(\beta_q) - P_{TDE}^{\alpha_p}(\beta_q), 0), & GT_{\beta_q} = 1, \\ max(P_{TDE}^{\alpha_p}(\beta_q) - P(\beta_q), 0), & GT_{\beta_q} = 0, \end{cases} \quad (10)$$

we want the affordance probablity change direction is right according to the GT affordance labels. Concretely, for an instance with the labeled causal relation between $[\alpha_p, \beta_q]$, if the label $GT_{\beta_q} = 1$, we expect the probability change $P(\beta_q) - P_{TDE}^{\alpha_p}(\beta_q)$ to be larger after eliminating the effect of $\alpha_p$. Because without the effect of $\alpha_p$, the probability of $\beta_q$ should be contrary to the fact ($GT_{\beta_q} = 1$), i.e., $P_{TDE}^{\alpha_p}(\beta_q)$ should be much smaller than $P(\beta_q)$. Similarly, if its $GT_{\beta_q} = 0$, we expect $P_{TDE}^{\alpha_p}(\beta_q)$

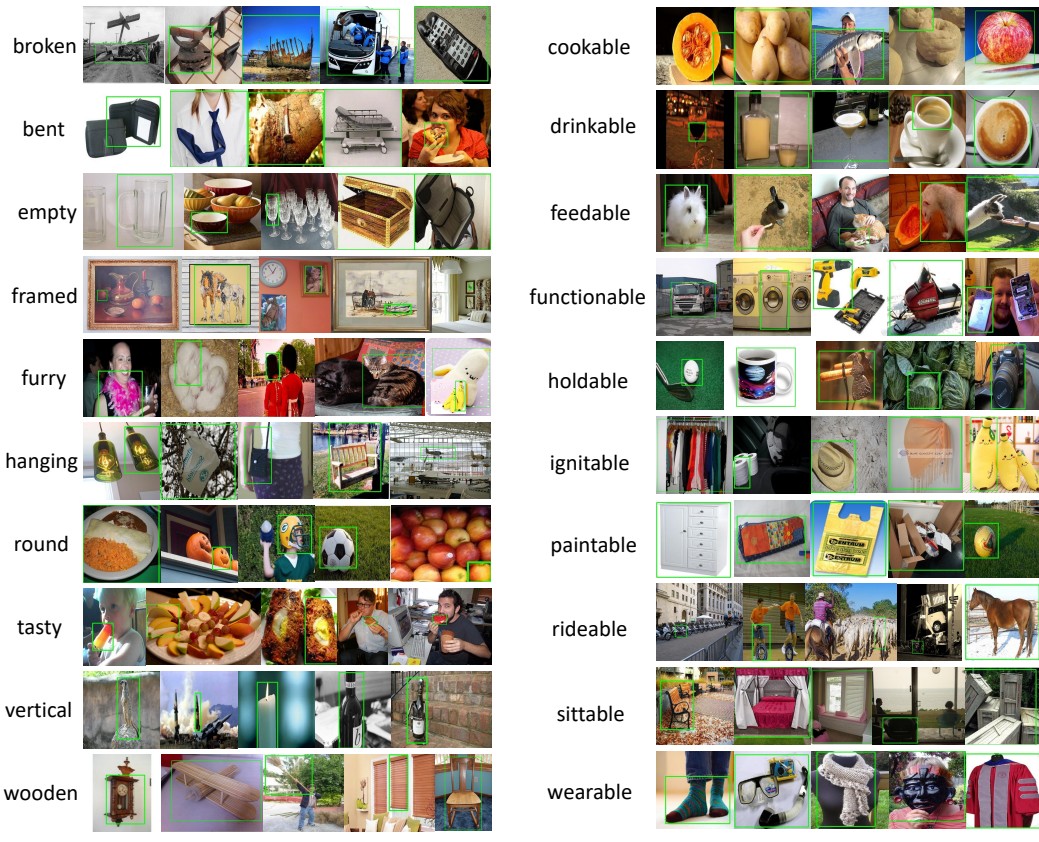

(a) attributes       (b) affordances

Figure 25: More OCL samples of attributes and affordances

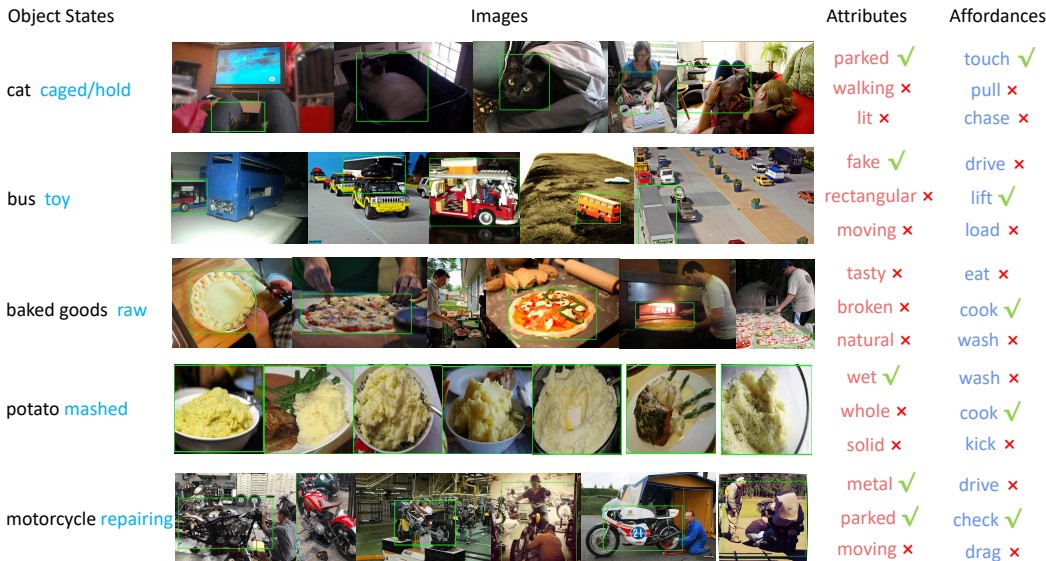

Figure 26: More OCL samples. We present objects in different states, together with their key attributes and affordances.

should be much larger than $P(\beta_q)$. That said, $P_{TDE}^{\alpha_p}(\beta_q) - P(\beta_q)$ should be large enough to follow the causal relation. Furthermore, in TDE loss, the setting follows this thought too.

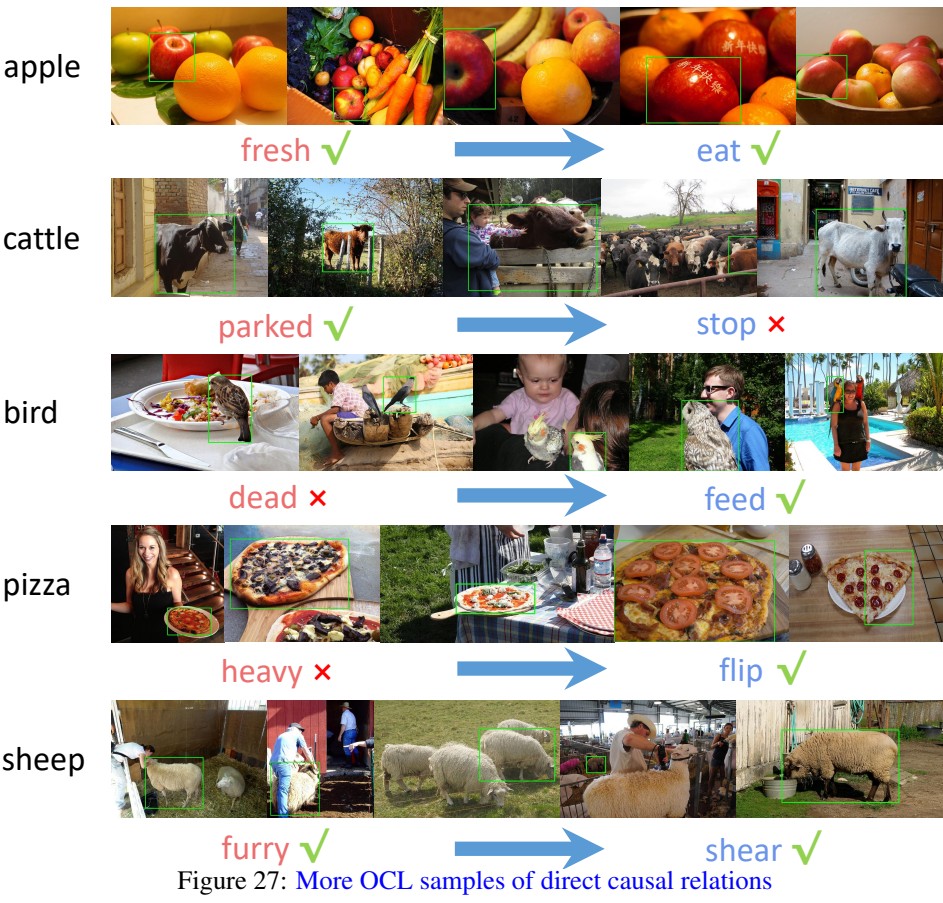

Figure 27: More OCL samples of direct causal relations

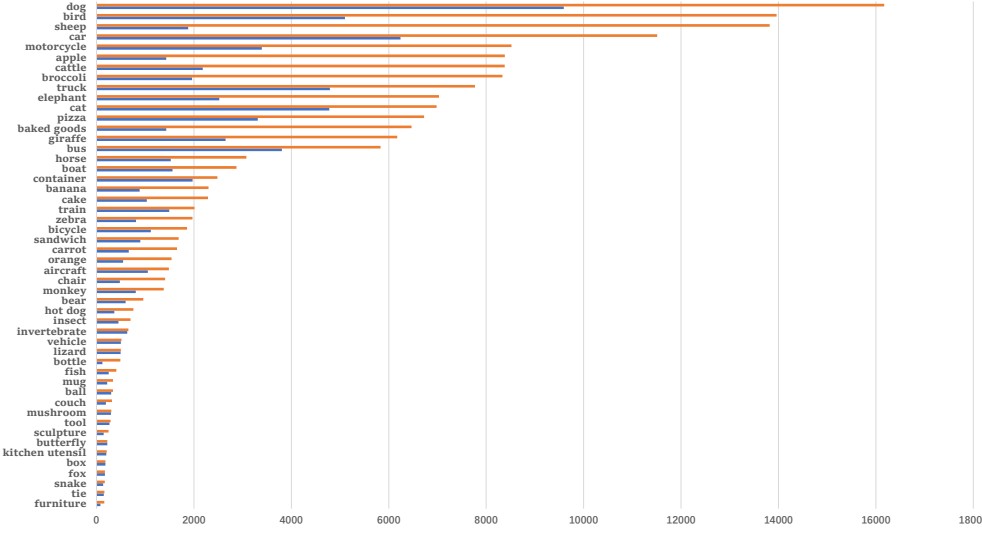

Figure 28: Counts of object categories.

Moreover, in $\alpha$-$\beta$-TDE, $P(\alpha_p = GT_{\alpha_p})$ means that the probabilities of predicted $P(\alpha_p)$ accords with the label $GT_{\alpha_p}$. In detail, if $GT_{\alpha_p} = 1$, then $P(\alpha_p = GT_{\alpha_p}) = P(\alpha_p)$; if $GT_{\alpha_p} = 0$, then $P(\alpha_p = GT_{\alpha_p}) = 1 - P(\alpha_p)$, where $P(\alpha_p)$ is the model prediction. $P(\beta_q = GT_{\beta_q})$ is similar. That said, if $GT_{\beta_q} = 1$, then $P(\alpha_p = GT_{\beta_q}) = P(\beta_q)$; if $GT_{\beta_q} = 0$, then $P(\beta_q = GT_{\beta_q}) = 1 - P(\beta_q)$. Finally, $TDE^{\alpha_p}_{\beta_q} * P(\alpha_p = GT_{\alpha_p}) * P(\beta_q = GT_{\beta_q})$ indicates the unified performance of attribute, affordance recognition and causal inference.

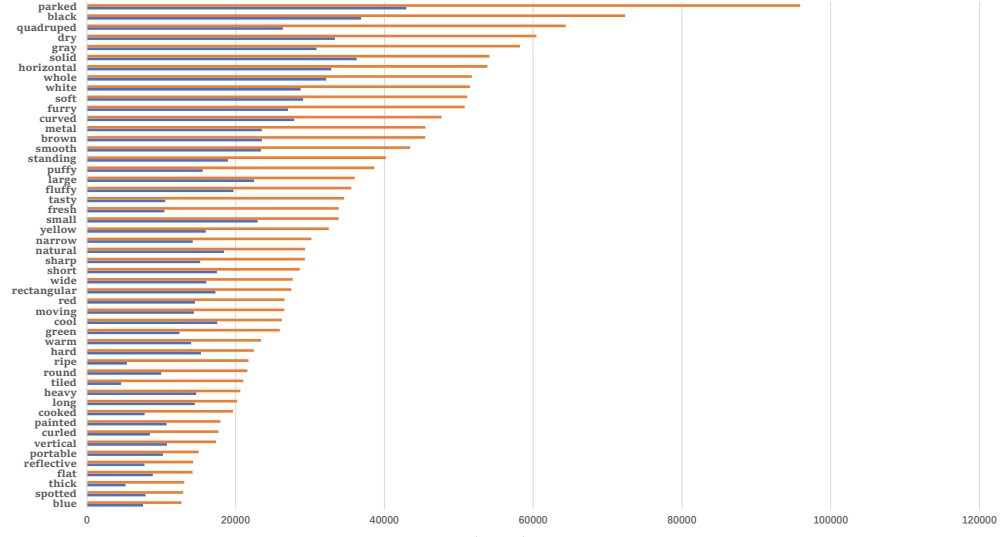

Figure 29: Counts of attribute classes.

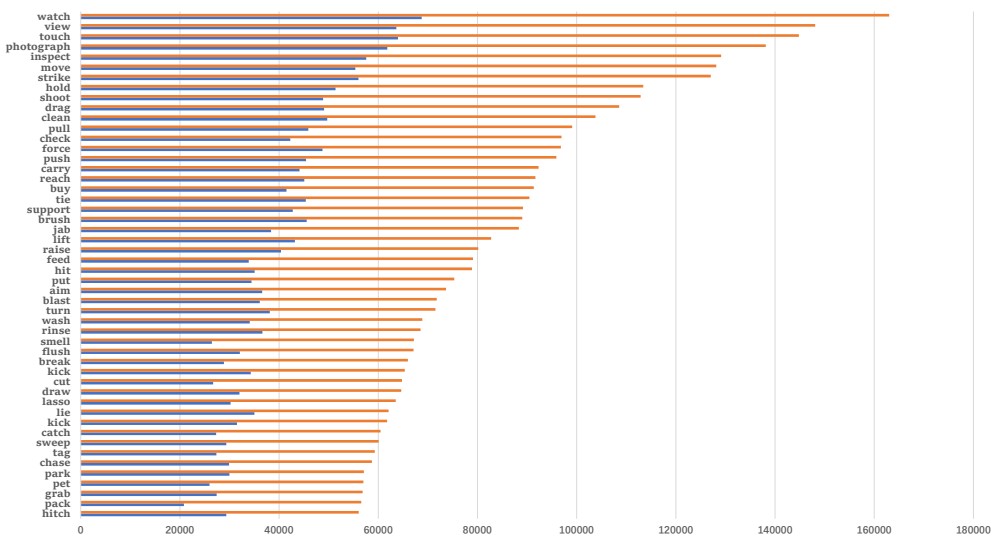

Figure 30: Counts of affordance classes.

For attribute/affordance recognition only, all methods adopt the labels to learn knowledge from the data. Moreover, in the evaluation of causal relation, only the "w/ $L_{TDE}$" adopts the causal relation labels. We hope the models can self-learn to mine and follow the intrinsic causalities. Thus, we design the TDE operation to evaluate this ability. Some works Wang et al. (2020a); Tang et al. (2020a;b) also try to marry supervised deep learning and causal inference which are similar to our OCRN method. Besides, current techniques especially the deep learning methods perform unsatisfying in our OCL task, and all show poor ability to capture the causality. We think our main aim and contribution here is to reveal this important problem via our OCL dataset instead of directly addressing the task we propose. A lot of works should be done to prompt this field which we hope our OCL can support.

## A.5 BASELINE DETAILS

We introduce the details of all baselines here:

**(1) Direct Mapping from Visual Feature (DM-V)**: feeding $f_I$ into MLP-Sigmoids to predict $P_\alpha, P_\beta$. Each $\alpha$ and $\beta$ class owns customized MLP followed by LayerNorm Ba et al. (2016) to generate class-specific feature and shares a same MLP-Sigmoid in classification.

**(2) DM from Attribute (DM-At)**: training an $\alpha$ classifier with $f_I$ same with DM-V, but use the concatenated representation of attributes as $f_\alpha$ to train the $\beta$ classifier.

**(3) DM from Attribute and Object (DM-AtO)**: training an $\alpha$ classifier with $f_I$ same with DM-V, but use the concatenated representation of attributes $f_\alpha$ and objects $f_o$ to train the $\beta$ classifier.

**(4) DM from Linguistic Representation (DM-L)**: replace the input representation $f_I$ of DM-V with linguistic feature $f_L$, which is the expectation of Bert (Devlin et al., 2018) of category names w.r.t $P(O_i|I)$.

**(5) Multi-Modality (MM)**: mapping $f_I$ to the semantic space via minimizing the distance to its $f_L$. The multi-modal aligned $f_I$ is fed to MLP-Sigmoids to predict $P_\alpha, P_\beta$.

**(6) Markov Logic Network (MLN-GT)** (Richardson & Domingos, 2006): adopt MLN to model the $\alpha - \beta$ relations following Zhu et al. (2014). After training on OCL, we infer $\beta$ with **GT** $\alpha$ to estimate its *performance upper bound*.

**(7) Linguistic Correlation (LingCorr)**: measure the correlation between object and $\alpha/\beta$ classes via their Bert (Devlin et al., 2018) cosine similarity. $P_\alpha, P_\beta$ are given by multiplying $P(O|I)$ to correlation matrices.

**(8) Kernelized Probabilistic Matrix Factorization (KPMF) (Zhou et al., 2012)**: calculate the Softmax normalized cosine similarity between each testing instance and all training samples as weights. Then $P_\alpha$ or $P_\beta$ is generated as the weighted sum of GT $\alpha$ or $\beta$ of training samples.

**(9) A&B Lookup**: return the expectation of category-level attribute or affordance vectors $A_i, B_i$ w.r.t $P(O_i|I)$.

**(10) Attention**: map the $A$ and $f_I$ into a shared latent space then use Sigmoid to convert $f_I$ into attention to generate instantialized $f_\alpha$ followed by MLP-Sigmoids to predict $\alpha$. As for $\beta$, we use the concatenated representation of $f_I$ and $f_\alpha$ to generate attenion on $B$.

**(11) Hierarchical Mapping (HMa)**: first mapping $f_I$ to category-level attribute or affordance space by an MLP supervised by GT $A$ or $B$. Then the mapped features are fed to an MLP-Sigmoid to predict $P_\alpha$ or $P_\beta$.

**(12) Ngram** (Lin et al., 2012): Ngram is adopted to retrieve the relevance between $\alpha$ and $\beta$ and generate an association matrix $M_{\alpha-\beta}$. Then we multiply DM predicted $P_\alpha$ with $M_{\alpha-\beta}$ to estimate $P_\beta$.

Besides, the TDE calculation needs feature zero-masking to eliminate the effect of specific attributes (Tang et al., 2020b). Thus, some baselines (DM-V, DM-L, MM, LingCorr, KPMF, $A\&B$-Lookup, HMa) cannot be evaluated with TDE. The other methods (DM-At, DM-AtO, Attention, OCRN) follow the same TDE calculation (feature masking). Two unique cases are Ngram and MLN-GT. Ngram uses attribute probabilities to infer affordance. Thus, we randomize the specific attribute probabilities for Ngram to operate the TDE calculation. And MLN-GT must use GT attribute labels to distinguish the "positive" and "negative" causes and then reason out the effect affordance. Thus, in TDE, we directly eliminate its corresponding attribute input.

## A.6  DETAILED RESULT ANALYSIS

### A.6.1  DETAILED ATTRIBUTE AND AFFORDANCE PERFORMANCES

We compute and analysis the performance (AP) of OCRN on each attribute or affordance class in Fig. 31 and Fig. 32, which suggest that visually abstract concepts like `fake` are more difficult to model than concrete ones like `metal`, `breakable`. The performance of attribute classes is lower than affordance classes. This is mainly because the attributes have more diversities. Thus the *positive* instances of each attribute class are **less** than the affordance class.

### A.6.2  ATTRIBUTE AND AFFORDANCE RECOGNITION GIVEN DETECTED BOXES

Though OCL is a high-level concept learning task with object boxes as inputs, we can also consider object detection in evaluation for practical applications. We adopt Swin transformer (Swin) (Liu et al., 2021) as the detector. It is pretrained on COCO (Lin et al., 2014) and finetuned on OCL train

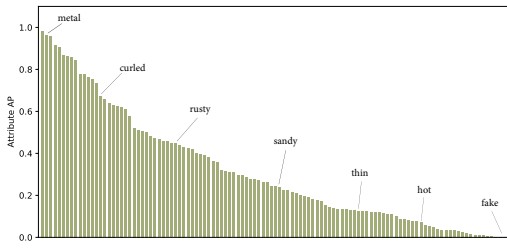 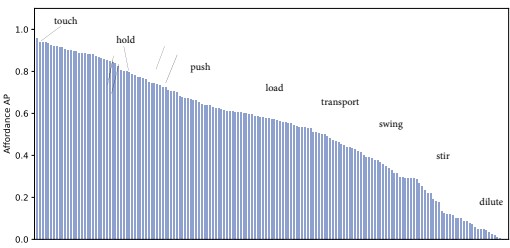

Figure 31: AP of attribute classes.   Figure 32: AP of affordance classes.

set with GT boxes of 381 categories. On OCL test set, it achieves 22.9 $AP_{50}$ on object detection. Subsequently, it will provide detected box $b_o$ for all models in inference. We can consider the detection effect in the attribute and affordance recognition metric to build a more strict criterion. Namely, all *false positive* detections (IoU<0.3 with refer to GT boxes) as the *false positives* of $\alpha$ and $\beta$ recognition too. Moreover, TDE calculation needs to construct the counterfactual of an object instance. If the inaccurately detected object box shifts according to the GT box, it is difficult to know whether the counterfactual comes from the attribute masking or visual content change, using the corresponding attribute-affordance causal relation labels of this GT box. Thus, considering the unique property of causal inference different from common recognition, here we do not report the TDE score. Tab. 5 shows the results given detected boxes. Due to the more strict criterion and detection quality, the performances of all methods degrade greatly. But OCRN still holds the superiority on two tracks.

| Method | $\alpha$ | $\beta$ |
|---|---|---|
| DM-V | 7.4 | 11.0 |
| DM-L | 4.6 | 9.1 |
| MM | 5.4 | 9.9 |
| LingCorr | 1.7 | 5.6 |
| KPMF | 6.4 | 10.5 |
| $A\&B$-Lookup | 4.1 | 5.8 |
| HMa | 6.5 | 10.9 |
| DM-At | 6.8 | 10.5 |
| DM-AtO | 6.6 | 10.8 |
| Ngram | 5.1 | 10.2 |
| MLN-GT | - | - |
| Attention | 5.5 | 10.1 |
| OCRN | **7.9** | **11.3** |

Table 5: Attribute and affordance recognition results given detected boxes from Swin Transformer (Liu et al., 2021).

### A.6.3 OCL-BASED IMAGE RETRIEVAL

We visualize the OCL reasoning performance by retrieval the top-score instances with OCRN. Some results are shown in Fig. 34. The model can correctly retrieve the related images, especially on some common concepts e.g. columnar, sit.

### A.7 APPLICATION ON HUMAN-OBJECT INTERACTION (HOI) DETECTION

To further verify the generalization ability, we apply OCL to Human-Object Interaction (HOI) detection and help HOI methods boost their performances.

HOI depicts the actions performed upon objects by humans. Usually, an object has multi-affordance, i.e., a person can perform different actions upon it. But in an image, just one or several actions/affordances are usually happening/**activated**. Without object knowledge, previous methods (Li et al., 2019; Gao et al., 2018; Li et al., 2020a) can find the activated affordances from hundreds of actions (Chao et al., 2018). For example, for each human-object pair in HICO-DET (Chao et al., 2018), a model has to select one or several actions from the defined 116 actions. With OCL, things

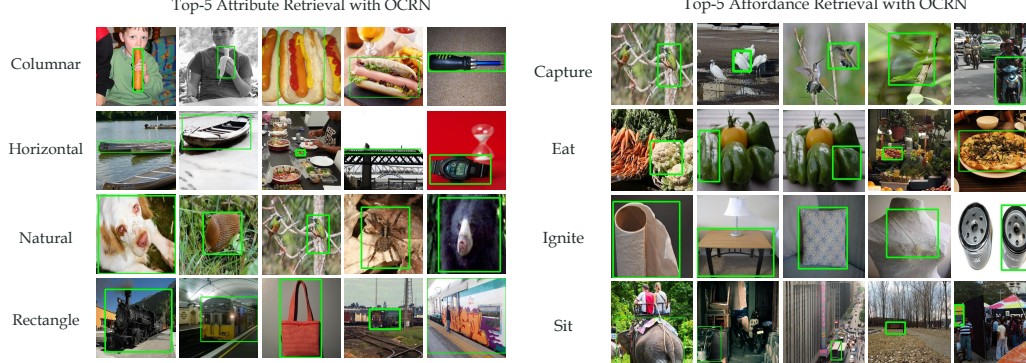

Figure 33: Top-5 attribute retrievals on the OCL test set.

Figure 34: Top-5 affordance retrievals on the OCL test set.

are different. OCL covers many actions, so we can use OCRN to infer $P_\beta$ of an object to narrow the solution space. Thus, we propose two ways:

(1) **OCL Filtering**: We use $P_\beta$ to narrow the action space with a threshold $\gamma$ and generate $P_\beta^\gamma$. Affordances with probabilities higher than $\gamma$ are kept and others are set to *zero* ($\gamma = 0.5$). Then, HOI model only needs to predict in a narrowed action space. In practice, we multiply the prediction $P_{HOI}$ from HOI model with $P_\beta^\gamma$ element-wisely to obtain the final prediction $P'_{HOI} = P_{HOI} * P_\beta^\gamma$.

(2) **Human-as-Probe**: Another more straightforward way is to predict HOI via OCL directly. We treat the human paired with the object as a **probe**. Assuming the human feature is $f_h$ and human-object spatial configuration feature is $f_{sp}$ (from Li et al. (2019); Gao et al. (2018)). As $P_\beta$ indicates all possible affordances, the ongoing actions can be seen as the **instantiation** of $P_\beta$, i.e., they are activated by the "probe" $f_h$ and $f_{sp}$. So we use $f_h$ and $f_{sp}$ to generate an attention $A_{h+sp}$ via MLP-Sigmoid. Then we operate $P_\beta * A_{h+sp}$ and late fusion to get the final prediction $P'_{HOI} = (P_\beta * A_{h+sp} + P_{HOI})/2$.

Concretely, we use OCRN to enhance HOI detection models (iCAN (Gao et al., 2018), TIN (Li et al., 2019), IDN (Li et al., 2020a)) on HICO-DET (Chao et al., 2018). As OCL merely contains 15 object categories in HICO-DET (Chao et al., 2018), the rest 65 object categories are **unseen**. We embed OCRN into three HOI models according to OCL filtering and Human-as-Probe (Appendix Sec. A.7), and the public model checkpoints of Gao et al. (2018); Li et al. (2019; 2020a) are used.

The results are shown in Tab. 6. With OCL filtering, iCAN (Gao et al., 2018), TIN (Li et al., 2019), and IDN (Li et al., 2020a) achieve a gain of mAP by 0.65%, 0.90%, and 0.77% respectively. The Human-as-Probe is more suitable for HOI detection and contributes a performance boost of 1.50%, 1.46%, and 0.98% to three models. These strongly verify the efficacy and generalization ability of OCL.

| Method | Full | Rare | Non-Rare |
|---|---|---|---|
| iCAN | 14.84 | 10.45 | 16.15 |
| iCAN+Filtering | 15.49 | 8.76 | 17.50 |
| iCAN+Probe | **16.34** | **11.66** | **17.74** |
| TIN | 17.03 | 13.42 | 18.11 |
| TIN+Filtering | 17.93 | 13.79 | 19.17 |
| TIN+Probe | **18.49** | **15.02** | **19.58** |
| IDN | 23.36 | 22.47 | 23.63 |
| IDN+Filtering | 24.13 | 23.74 | 24.24 |
| IDN+Probe | **24.34** | **24.03** | **24.43** |

Table 6: Results of HOI detection (using detected object boxes).

## A.8 COMPARISON ON DEBIASING

The motivation of the OCRN is to follow the prior knowledge of the three levels of objects with a deep learning-based causal graph model, to pursue the object understanding beyond the common direct mapping from pixels to labels. To avoid the bias estimation such as in the **Simpson's paradox** Pearl et al. (2016). Thus, we use intervention to deconfound the confounder *category* and exclusive the possible spurious bias and correlation imported bias from imbalanced object categories. Overall, we propose our OCRN in a causal inference perspective instead of the pure classification viewpoint, which also suits our causal graphical model well. Similar cases are also proposed in recent works like Wang et al. (2020a); Tang et al. (2020a;b). Moreover, to better compare our method with the common debiasing methods, we further conduct the experiments as follows.

We regard $\alpha, \beta$ recognition as multiple independent binary classification tasks and implement some methods introduced in Wang et al. (2020b) on our strong baseline DM-V to reduce bias from object categories. We use **mean bias amplification** (**Amp**) in Zhao et al. (2017) as bias evaluation metric: small Amp means model suffers less from data category bias. The test results are shown in Tab. 7. The proposed OCRN has comparable or smaller bias amplification than the variants of DM-V since our model follows the causal graph and exploits the tools of causal inference, while most methods for category bias are from the view of classification.

| Model | Test Inference | $\alpha$ Amp. | $\beta$ Amp. |
|---|---|---|---|
| OCRN | $\mathrm{argmax}_y P(y|x)$ | **0.127** | **0.112** |
| DM-V + Joint ND-way Softmax | $\mathrm{argmax}_y \max_d P_{te}(y,d|x)$ | 0.151 | 0.158 |
| DM-V + Joint ND-way Softmax | $\mathrm{argmax}_y \sum_d P_{te}(y,d|x)$ | 0.148 | 0.154 |
| DM-V + N-way classifier per domain | $\mathrm{argmax}_y P_{te}(y|d^*,x)$ | 0.135 | **0.112** |
| DM-V + N-way classifier per domain | $\mathrm{argmax}_y \sum_d s(y,d,x)$ | 0.147 | 0.145 |

Table 7: Comparison with debiasing models.

## A.9 ABOUT STATES

We did not use object states in our model because there is a **compositional zero-shot problem** for object-state pairs, i.e., there can be **unseen** states in real-world data. Differently, affordances are more general. The models explicitly incorporating object states will fail to generalize to these zero-shot states and it adds to the object category bias. In experiments, the state supervision during training would indeed slightly improve the affordance recognition performance, since instances in the **same state** lies in a tight cluster in affordance label space. But this will hurt the TDE performance greatly.

## A.10 DISCUSSION ABOUT CAUSALITY AND CAUSAL GRAPH

Annotating causality in the real-world is extremely difficult. In data annotation, we have met numerous ambiguities and difficulties to confirm the "right" causal relations. To address these challenges, we follow the following principles: (1) Firstly, we only emphasize on **clear** and **strong** causal relations via crowdsourcing, but omit the vague ones. (2) Second, we take an object **affordance-centric** viewpoint to look at the possible causal relations. (3) We would rather discard than condone the controversial situation in annotation. (4) We only focus on the simple relations between **one** attribute and **one** affordance, instead of the very complex compositions of multiple attributes and affordances which are almost impossible to annotate. Therefore, we finally find that we can label **a very small percentage** of all edges with the whole causal graph consisting of so many nodes (category, attributes, affordances, contexts, etc.) while keeping the quality.

Our causal graph follows the human priors from our experts and crowdsourcing annotators. Some previous works also follow this before designing the method, such as Zhu et al. (2014). From the viewpoint of causal discovery Pearl et al. (2016); Tang et al. (2020a;b); Wang et al. (2020a), the above arcs (e.g., the inverted arc from attribute to category in the causal graph directed acyclic graph, DAG) are indeed possible. However, here, we mainly study the object concept learning problem, especially attribute and affordance learning. Thus, from the perspective of affordance learning, we think the arcs from category to attribute and affordance is more vital and meaningful to us.

More specifically: (1) The causal graph follows the nature of **human-like recognition**. Given a new object, we refer to the category level commonsense first, then a specific instantiation. Considering a simple case, when a human wants to find an apple to eat, he will pick up a fresh one instead of a bad one based on the recognition of object category and attribute. (2) Definitely, the causal graph reflects the **data collection pipeline**. We first select object categories before collecting images and complementing instances. After that, category-level $A, B$ and instance-level $\alpha, \beta$ are annotated. (3) Different arcs are mainly attributed to **task context**, e.g., we have "image to label" in image recognition task, but "label to image" in image generation task. In OCL, the taxonomy viewpoint (how we define an object due to its attribute or affordance, e.g., a container is a container because it can be used to hold or transport something) lies out of the area of our concern. Instead, we consider more on the **physical** side of learning the property and functionality of an object for **intelligent agent/robot**. It will have a larger impact on boosting downstream tasks like robot manipulation.

## A.11 DETAILED LISTS

The detailed object categories, states, attributes, and affordances are listed as follows.

### A.11.1 OBJECT CATEGORY LIST

abaya, academic gown, accordion, acorn, aircraft, alpaca, ambulance, apple, armadillo, artichoke, axe, backpack, bagel, baked goods, balance beam, ball, balloon, banana, banjo, barbell, barrel, bathtub, beach wagon, beaker, bear, bed, bee, bell pepper, bench, bib, bicycle, billiard table, binoculars, bird, bison, boat, bookcase, boot, bottle, bottle opener, box, brace, brassiere, bread, broccoli, broom, bubble, buckeye, bull, burrito, bus, butterfly, cabbage, cake, camel, camera, can opener, candle, cannon, car, cardigan, carnivore, carrot, cart, cat, cattle, cavy, cello, chain, chainsaw, chair, cheetah, chest of drawers, chicken, chime, clock, clog, closet, coat, cocktail shaker, coffee, computer keyboard, computer mouse, conch, container, corn, couch, cowboy hat, crab, crocodile, crossword puzzle, crutch, curtain, custard apple, desk, dessert, dhole, diaper, digital clock, dinosaur, dishwasher, dog, door, dough, dragonfly, drill, drum, drumstick, duck, dugong, dumbbell, eagle, eggnog, elephant, envelope, eraser, face powder, filing cabinet, fish, flower, flowerpot, flute, football helmet, footwear, fountain, fox, french horn, frog, frying pan, furniture, gazelle, giraffe, glove, goat, goblet, golf ball, golf cart, goose, gown, guacamole, guitar, hair dryer, hair slide, hair spray, hamburger, hammer, hamster, handgun, handkerchief, harmonica, harp, hartebeest, hat, hay, heater, hedgehog, helmet, hen, hippopotamus, home appliance, honeycomb, horizontal bar, horse, hot dog, hyena, ibex, ice cream, impala, insect, invertebrate, ipod, isopod, jaguar, jeans, jellyfish, jinrikisha, joystick, jug, kimono, kitchen utensil, knee pad, knife, knot, koala, lab coat, ladle, lamp, lampshade, lantern, laptop, lemon, letter opener, lighthouse, limousine, lion, lipstick, lizard, loudspeaker, loupe, madagascar cat, mailbox, maracas, marimba, mask, matchstick, measuring cup, meat loaf, megalith, microphone, microwave oven, military uniform, minibus, miniskirt, mirror, missile, mixing bowl, mobile home, mobile phone, modem, mongoose, monkey, mortarboard, motorcycle, mug, mushroom, nail, necklace, nipple, oar, oboe, ocarina, oil filter, orange, otter, oven, owl, oxygen mask, packet, paintbrush, panda, panpipe, paper towel, parachute, parking meter, pen, pencil case, pencil sharpener, penguin, petri dish, piano, picket fence, pig, pillow, pinwheel, pizza, plastic bag, plate, polar bear, pomegranate, porch, potato, pretzel, printer, rabbit, radio telescope, rain barrel, red panda, reel, refrigerator, remote control, rifle, rocking chair, rugby ball, safety pin, sandal, sandwich, sarong, saxophone, scarf, scoreboard, scorpion, screw, screwdriver, sculpture, sea lion, seat belt, sewing machine, shark, sheep, shield, shirt, shoji, shower, shower cap, ski, ski mask, sleeping bag, slide rule, snail, snake, snorkel, snowmobile, snowplow, soap dispenser, sock, solar dish, sombrero, spatula, spider, squash, squirrel, starfish, steel drum, stethoscope, strawberry, stretcher, submarine, suit, sunglasses, sunscreen, swan, swimwear, swing, syringe, tank, taxi, teapot, teddy bear, telephone, tennis ball, tent, tick, tie, timber wolf, toaster, toilet paper, tool, torch, towel, tower, traffic light, train, tripod, trombone, truck, trumpet, turnstile, turtle, umbrella, unicycle, vase, vegetable, vehicle, vending machine, vestment, violin, volleyball, vulture, waffle iron, wall clock, wallaby, wallet, wardrobe, warthog, washing machine, watch, weapon, whale, wheel, whistle, wild boar, windmill, window shade, wine, wok, wooden spoon, wool, worm, yawl, yurt, zebra.

### A.11.2 Object Attribute List

crumpled, ceramic, cold, curled, furry, black, wet, orange, brown, yellow, striped, cool, gray, leather, large, wooden, small, soft, round, old, portable, fluffy, hard, horn, messy, heavy, blue, purple, closed, new, red, thin, full, vertical, strong, dry, spotted, quadruped, whole, sharp, long, fake, open, toy, plastic, white, columnar, empty, flat, cloth, warm, leashed, solid, smooth, worn, rectangular, bipedal, tasty, curved, pink, hot, digital, electric, fresh, horizontal, short, natural, metal, cooked, green, folded, broken, bent, sliced, thick, wide, narrow, arched, puffy, cream, stone, cement, marble, floral, glass, water, rubber, brick, sandy, plaid, paper, checkered, parked, moving, melted, lit, wearing, framed, stacked, tiled, standing, hanging, sitting, walking, sleeping, flying, dead, ripe, in the picture, reflective, grassy, leafy, painted, rusty.

### A.11.3 Object State List

**(1) abaya**: common, damaged, special material. **(2) academic gown**: common, damaged, special material. **(3) accordion**: common, in picture. **(4) acorn**: common, rotten. **(5) aircraft**: broken, flying, in factory, model, on ground, on show, part, toy. **(6) alpaca**: caged, dead, fighting, hugging, lying, milking, picture, playing, running, shearing, standing, statue, toy, walking. **(7) ambulance**: broken, building, food, full, load on vehicle, model, moving, parked, part, picture, repairing, stained, wrapped. **(8) apple**: eating, fake, holding, model, picture, piece, rotten, sliced, toy, whole. **(9) armadillo**: dead, model, moving, resting, with shell. **(10) artichoke**: common. **(11) axe**: common, model, toy. **(12) backpack**: normal canvas. **(13) bagel**: eating, fake, holding, picture, raw, sliced, whole. **(14) baked goods**: eating, fake, holding, picture, raw, sliced, whole. **(15) balance beam**: common. **(16) ball**: flying large ball, flying small ball, holding large ball, holding small ball, kicking, playing large ball, playing small ball. **(17) balloon**: flying, normal landed. **(18) banana**: cooked, immature, model, on tree, peeled, picture, piece, rotten, sliced, toy, unripe. **(19) banjo**: common, playing. **(20) barbell**: common. **(21) barrel**: common, full. **(22) bathtub**: common. **(23) beach wagon**: broken, building, food, full, load on vehicle, model, moving, parked, part, picture, repairing, stained, wrapped. **(24) beaker**: ceramic glass, wood plastic. **(25) bear**: caged, dead, hugging, model, picture, running, standing, swimming, tied, title page, trapped. **(26) bed**: common. **(27) bee**: flying, on some places. **(28) bell pepper**: common. **(29) bench**: common. **(30) bib**: common. **(31) bicycle**: broken, fake, holding, load on vehicle, model, moving, parked, picture, riding, stained. **(32) billiard table**: common. **(33) binoculars**: fixed, portable. **(34) bird**: caged, dead, flying, high place, hold, meat, model, perched, picture, swimming. **(35) bison**: caged, dead, eating, lying, model, moving, picture, toy. **(36) boat**: model, moving, parked, stacked, stained. **(37) bookcase**: common, empty. **(38) boot**: common, fake, wearing. **(39) bottle**: glass, glass broken, metal, plastic. **(40) bottle opener**: with drill. **(41) box**: metal closed, metal locked, metal open, paper closed, paper open, wood closed, wood open. **(42) brace**: common. **(43) brassiere**: common. **(44) bread**: eating, fake, picture, raw, sliced, whole. **(45) broccoli**: cooked, fake, raw, rotten. **(46) broom**: common, fractured. **(47) bubble**: plastic, soap. **(48) buckeye**: with shell. **(49) bull**: caged, dead, eating, lying, model, moving, picture, toy. **(50) burrito**: eating, fake, holding, picture, raw, sliced, whole. **(51) bus**: broken, building, food, full, load on vehicle, model, moving, parked, picture, repairing, stained, toy, wrapped. **(52) butterfly**: flying, on hand, on some places. **(53) cabbage**: common. **(54) cake**: eating, fake, picture, raw, sliced, whole. **(55) camel**: sitting, standing, walking. **(56) camera**: common. **(57) can opener**: hidden blade, with blade, with drill. **(58) candle**: common, lit. **(59) cannon**: common, model. **(60) car**: broken, building, food, full, load on vehicle, model, moving, parked, part, picture, repairing, stained, wrapped. **(61) cardigan**: common, damaged, special material. **(62) carnivore**: caged, moving, sitting, toy. **(63) carrot**: cooked, dirty, fake, rotten, sliced, whole. **(64) cart**: metal, plastic, wood. **(65) cat**: caged, fake, holding, lying, moving, picture, standing, toy. **(66) cattle**: caged, dead, fighting, hugging, lying, milking, picture, playing, running, shearing, standing, statue, toy, walking. **(67) cavy**: eating, holding, playing, resting. **(68) cello**: common. **(69) chain**: thick, thin. **(70) chainsaw**: common. **(71) chair**: broken, burnable, fragile, fragile and burnable, solid. **(72) cheetah**: caged, moving, sitting, toy. **(73) chest of drawers**: wood. **(74) chicken**: caged, dead, flying, high place, hold, meat, model, perched, picture, swimming. **(75) chime**: large, small, wood plastic. **(76) clock**: broken, common. **(77) clog**: boat, common, wearing. **(78) closet**: common. **(79) coat**: common, damaged, special material. **(80) cocktail shaker**: common. **(81) coffee**: common. **(82) computer keyboard**: common. **(83) computer mouse**: broken, common. **(84) conch**: alive, shell. **(85) container**: closed, full, locked, open, with holes. **(86) corn**: kernels, rotten, whole, with husk. **(87) couch**: common. **(88) cowboy hat**: common. **(89) crab**:

alive, cooked. **(90) crocodile**: common. **(91) crossword puzzle**: common. **(92) crutch**: common. **(93) curtain**: common. **(94) custard apple**: common. **(95) desk**: common. **(96) dessert**: common, half eaten, hold by human. **(97) dhole**: caged, moving, sitting, toy. **(98) diaper**: common, hanged, wearing. **(99) digital clock**: broken, common, hold by human, projected in screen. **(100) dinosaur**: fake, fossil, sculpture. **(101) dishwasher**: broken, closed, fixing, opened, playing with human pet, uninstalled, using. **(102) dog**: caged, in boat, in car, jumping, lying, model, moving, picture, playing, stop, tied, toy. **(103) door**: close, open, open with human. **(104) dough**: common. **(105) dragonfly**: on some places. **(106) drill**: occupied, placed. **(107) drum**: occupied, placed. **(108) drumstick**: occupied, placed. **(109) duck**: caged, dead, flying, high place, hold, meat, model, perched, picture, swimming. **(110) dugong**: eating, swimming. **(111) dumbbell**: occupied, placed. **(112) eagle**: flying, in high place, on ground. **(113) eggnog**: common, half eaten, hold by human. **(114) elephant**: caged, in water, lying, on show, picture, playing, ridden, sitting, standing, statue, toy, trap, walking. **(115) envelope**: common, hold by human. **(116) eraser**: common, run out. **(117) face powder**: common, on brush. **(118) filing cabinet**: close, open. **(119) fish**: as food, caught by human, normal(swimming). **(120) flower**: common. **(121) flowerpot**: common. **(122) flute**: common, playing. **(123) football helmet**: common, wearing. **(124) footwear**: common, underwear swimwear, wearing. **(125) fountain**: close, operative. **(126) fox**: caged, moving, sitting, toy. **(127) french horn**: common. **(128) frog**: in hand, on ground, swimming. **(129) frying pan**: common, hold by human, using. **(130) furniture**: common, using. **(131) gazelle**: caged, dead, fighting, hugging, lying, milking, picture, playing, running, shearing, standing, statue, toy, walking. **(132) giraffe**: caged, dead, fighting, hugging, lying, milking, picture, playing, running, shearing, standing, statue, toy, walking. **(133) glove**: not wearing, wearing, weaving. **(134) goat**: caged, dead, fighting, hugging, lying, milking, picture, playing, running, shearing, standing, statue, toy, walking. **(135) goblet**: empty, full. **(136) golf ball**: building, common. **(137) golf cart**: broken, building, food, full, load on vehicle, model, moving, parked, part, picture, repairing, stained, wrapped. **(138) goose**: caged, dead, flying, high place, hold, meat, model, perched, picture, swimming. **(139) gown**: common, damaged, special material. **(140) guacamole**: pure, with food. **(141) guitar**: common, playing. **(142) hair dryer**: model, place, use. **(143) hair slide**: hold by human, place. **(144) hair spray**: building, hold, place. **(145) hamburger**: eating, fake, picture, raw, sliced, whole. **(146) hammer**: common. **(147) hamster**: holding, playing, resting eating. **(148) handgun**: common, hold by human. **(149) handkerchief**: hold by human, placed. **(150) harmonica**: place, playing. **(151) harp**: place, playing. **(152) hartebeest**: caged, dead, fighting, hugging, lying, milking, picture, playing, running, shearing, standing, statue, toy, walking. **(153) hat**: on hand, place, toy, wearing. **(154) hay**: common. **(155) heater**: closed, running. **(156) hedgehog**: common, eating. **(157) helmet**: place, wear. **(158) hen**: caged, dead, flying, high place, hold, meat, model, perched, picture, swimming. **(159) hippopotamus**: caged, in water, on ground. **(160) home appliance**: broken, placed, using. **(161) honeycomb**: common, processed food. **(162) horizontal bar**: common, grab or used in sports. **(163) horse**: lying, model, picture, pulling, riding, running, standing, statue, toy. **(164) hot dog**: eating, fake, picture, raw, sliced, whole. **(165) hyena**: caged, moving, sitting, toy. **(166) ibex**: caged, dead, fighting, hugging, lying, milking, picture, playing, running, shearing, standing, statue, toy, walking. **(167) ice cream**: hold, placed. **(168) impala**: caged, dead, fighting, hugging, lying, milking, picture, playing, running, shearing, standing, statue, toy, walking. **(169) insect**: caught in box,bag,cup, common, hold by human, on plant. **(170) invertebrate**: building, caught, common, food, fossil, in water, model toy. **(171) ipod**: on hand, place. **(172) isopod**: common, hold by human. **(173) jaguar**: caged, moving, sitting, toy. **(174) jeans**: common, wearing. **(175) jellyfish**: common. **(176) jinrikisha**: metal, wood plastic. **(177) joystick**: hold by human, place. **(178) jug**: common. **(179) kimono**: common, damaged, special material. **(180) kitchen utensil**: broken, common, contain something, heating. **(181) knee pad**: common, wearing. **(182) knife**: common, hold by human. **(183) knot**: common, noddles knot. **(184) koala**: common. **(185) lab coat**: common, damaged, special material. **(186) ladle**: contain something, hold by human, placed plastic or metal, placed wood. **(187) lamp**: common, uninstalled, using. **(188) lampshade**: common. **(189) lantern**: lighting, making, no light. **(190) laptop**: placed, using. **(191) lemon**: sliced, whole. **(192) letter opener**: hold by human, place. **(193) lighthouse**: common. **(194) limousine**: broken, building, food, full, load on vehicle, model, moving, parked, part, picture, repairing, stained, wrapped. **(195) lion**: caged, moving, sitting, toy. **(196) lipstick**: close, open, using. **(197) lizard**: dead, model, moving, resting, with shell. **(198) loudspeaker**: common, hold by human. **(199) loupe**: hold by human, place. **(200) madagascar cat**: caged, fake, holding, lying, moving, picture, standing, toy. **(201) mailbox**: steal, wooden. **(202) maracas**: common, using. **(203) marimba**: common, fake, using. **(204) mask**: common, wearing. **(205) matchstick**: common, in fire, in hand, used. **(206)**

**measuring cup**: ceramic glass, wood plastic. **(207) meat loaf**: eating, on table. **(208) megalith**: common. **(209) microphone**: common, using. **(210) microwave oven**: common, using. **(211) military uniform**: common, damaged, special material. **(212) minibus**: broken, building, food, full, load on vehicle, model, moving, parked, part, picture, repairing, stained, wrapped. **(213) miniskirt**: wearing. **(214) mirror**: rear view in car, rear view out car, street mirror. **(215) missile**: launched, unlaunched. **(216) mixing bowl**: common. **(217) mobile home**: close, moving, open, stop. **(218) mobile phone**: common, using. **(219) modem**: common, hold by human, stripped. **(220) mongoose**: common. **(221) monkey**: eating, fake, in cage, in house, lying, on tree rope shelf high place, pet by human, playing, resting, sitting, standing, swimming, walking running. **(222) mortarboard**: common, hold, thrown, wearing. **(223) motorcycle**: load on vehicle, model, moving, on show, picture, repairing, riding, stopped, toy. **(224) mug**: balloon, ceramic, glass, metal. **(225) mushroom**: common, cooked, fake, hold by human, possible poisonous. **(226) nail**: bent, common, half nailed in. **(227) necklace**: common. **(228) nipple**: with bottle, without bottle. **(229) oar**: fixed, portable. **(230) oboe**: common. **(231) ocarina**: common, fake, hold by human, playing. **(232) oil filter**: common. **(233) orange**: immature, model, peel, picture, rotten, sliced, whole. **(234) otter**: common, feed by human, hold by human, in water. **(235) oven**: open, pot like. **(236) owl**: caged, dead, flying, high place, hold, meat, model, perched, picture, swimming. **(237) oxygen mask**: common. **(238) packet**: closed, open. **(239) paintbrush**: common. **(240) panda**: eating resting, playing. **(241) panpipe**: common, hold by human, playing. **(242) paper towel**: common, torn. **(243) parachute**: common. **(244) parking meter**: box like, with pole. **(245) pen**: feather, metal, plastic. **(246) pencil case**: metal, soft plastic canvas. **(247) pencil sharpener**: embedded blade, rotating. **(248) penguin**: normal resting. **(249) petri dish**: empty, in use. **(250) piano**: broken, common, locked, model. **(251) picket fence**: normal wood. **(252) pig**: caged, resting. **(253) pillow**: common. **(254) pinwheel**: normal paper. **(255) pizza**: eating, fake, picture, raw, sliced, whole. **(256) plastic bag**: empty, filled. **(257) plate**: dirty, empty, with dish. **(258) polar bear**: caged, dead, hugging, model, picture, running, standing, swimming, tied, title page, trapped. **(259) pomegranate**: opened, rotten, seeds, whole. **(260) porch**: empty, with chair. **(261) potato**: mashed, whole raw. **(262) pretzel**: eating, fake, picture, raw, sliced, whole. **(263) printer**: large machine, mini portable, small desktop. **(264) rabbit**: caged, common, holding. **(265) radio telescope**: common. **(266) rain barrel**: robber, wood. **(267) red panda**: common, feed or hold by human. **(268) reel**: only reel itself, with rod, with rod and string. **(269) refrigerator**: closed, open full, open not full. **(270) remote control**: broken, common. **(271) rifle**: common. **(272) rocking chair**: wood. **(273) rugby ball**: alone, holding, model. **(274) safety pin**: architecture, broken, close, hang, hurt human, open. **(275) sandal**: common, wearing. **(276) sandwich**: eating, fake, picture, raw, sliced, whole. **(277) sarong**: placed, wearing. **(278) saxophone**: common. **(279) scarf**: on cat, placed, wearing. **(280) scoreboard**: installed, manual write, portable. **(281) scorpion**: common, hold by human, scorpion like bag. **(282) screw**: common, plug into something. **(283) screwdriver**: common, hold by human, in package, plug in something. **(284) sculpture**: common, in high place, in picture. **(285) sea lion**: in water, on ground. **(286) seat belt**: common. **(287) sewing machine**: off, on. **(288) shark**: caught by wire, dead on ground, normal swim. **(289) sheep**: caged, dead, fighting, hugging, lying, milking, picture, playing, running, shearing, standing, statue, toy, walking. **(290) shield**: common. **(291) shirt**: common, wearing. **(292) shoji**: close, open. **(293) shower**: off, on. **(294) shower cap**: on head, place. **(295) ski**: placed, using by human. **(296) ski mask**: common, wearing. **(297) sleeping bag**: empty flatten, human inside, rolled. **(298) slide rule**: common, hold by human. **(299) snail**: common, hold by human. **(300) snake**: common, hold by human, in water, toy. **(301) snorkel**: placed, wearing. **(302) snowmobile**: broken, building, food, full, load on vehicle, model, moving, parked, part, picture, repairing, stained, wrapped. **(303) snowplow**: broken, building, food, full, load on vehicle, model, moving, parked, part, picture, repairing, stained, wrapped. **(304) soap dispenser**: installed, portable. **(305) sock**: wearing. **(306) solar dish**: common, half installed. **(307) sombrero**: light, on back, place, wearing. **(308) spatula**: common, hold by human, toy. **(309) spider**: caught in box,bag,cup, common, hold by human, on plant. **(310) squash**: cooked meal, sliced, whole ripe, whole unripe. **(311) squirrel**: common, hold by human, toy. **(312) starfish**: hold, in water, on ground, on sale. **(313) steel drum**: occupied, placed. **(314) stethoscope**: common. **(315) strawberry**: common. **(316) stretcher**: empty, human on it, lift by human. **(317) submarine**: model, move, on ground, park. **(318) suit**: common, wearing. **(319) sunglasses**: on hand, placed, wearing. **(320) sunscreen**: hold by human, placed. **(321) swan**: caged, dead, flying, high place, hold, meat, model, perched, picture, swimming. **(322) swimwear**: placed, wearing. **(323) swing**: empty, human on it. **(324) syringe**: contain something, empty, hold by human, stick into something. **(325) tank**: firing, moving, stopped, toy. **(326) taxi**: broken, building, food, full, load on vehicle,

model, moving, parked, part, picture, repairing, stained, wrapped. **(327) teapot**: common. **(328) teddy bear**: common. **(329) telephone**: domestic, public. **(330) tennis ball**: common, damaged, unopened. **(331) tent**: common, using. **(332) tick**: common, hold by human, on plant. **(333) tie**: common, wearing. **(334) timber wolf**: caged, moving, sitting, toy. **(335) toaster**: common, using. **(336) toilet paper**: common, exhausted. **(337) tool**: common, damaged old, fake, using. **(338) torch**: electronic not work, firing, firing hold by human. **(339) towel**: common, using. **(340) tower**: common, portable. **(341) traffic light**: common, fixing. **(342) train**: food, model, moving, parked, picture, stained, toy. **(343) tripod**: common. **(344) trombone**: common, separated. **(345) truck**: broken, building, food, full, load on vehicle, model, moving, parked, part, picture, repairing, stained, wrapped. **(346) trumpet**: common. **(347) turnstile**: common. **(348) turtle**: in hand, in water, on ground. **(349) umbrella**: common, hold by human. **(350) unicycle**: common, riding. **(351) vase**: common. **(352) vegetable**: common, in hand, rotten, sliced cooked, unpicked. **(353) vehicle**: broken, building, food, full, load on vehicle, model, moving, parked, part, picture, repairing, stained, wrapped. **(354) vending machine**: off, on. **(355) vestment**: common, damaged, special material. **(356) violin**: common, playing. **(357) volleyball**: flying, holding, static. **(358) vulture**: flying, in high place, on ground. **(359) waffle iron**: broken, fake, part, picture, placed, using. **(360) wall clock**: common. **(361) wallaby**: eating, lying, running walking, standing. **(362) wallet**: common. **(363) wardrobe**: close, open. **(364) warthog**: eating, standing lying, walking running. **(365) washing machine**: close, in wild, open, using. **(366) watch**: common, wearing. **(367) weapon**: common, fake, firing, hold by human, using. **(368) whale**: common, dead. **(369) wheel**: damaged, holding, in water, sculpture, stop, uninstalled, using, using in water, working. **(370) whistle**: common. **(371) wild boar**: eating, lying, standing, walking running. **(372) windmill**: common. **(373) window shade**: put down, taken up. **(374) wine**: common, fake, pouring. **(375) wok**: common, hold by human, using. **(376) wooden spoon**: contain something, hold by human, placed wood. **(377) wool**: cloth, skein, toy. **(378) worm**: common, in water, microorganism. **(379) yawl**: model, moving, parked, stacked, stained. **(380) yurt**: close, fake, open, using. **(381) zebra**: caged, lying, model, picture, running, standing, statue, tied, toy, wild.

### A.11.4 OBJECT AFFORDANCE LIST

aim (point or cause to go (blows, weapons, or objects such as photographic equipment) towards), assemble (create by putting components or members together), bend (form a curve), bend (change direction), blast (fire a shot), blow (exhale hard), board (get on board of (trains, buses, ships, aircraft, etc.)), break (become separated into pieces or fragments), brush (rub with a brush, or as if with a brush), buy (obtain by purchase; acquire by means of a financial transaction), capture (capture as if by hunting, snaring, or trapping), carry (have with oneself; have on one's person), catch (attract and fix), catch (take hold of so as to seize or restrain or stop the motion of), chase (go after with the intent to catch), check (examine so as to determine accuracy, quality, or condition), clean (make clean by removing dirt, filth, or unwanted substances from), climb (go upward with gradual or continuous progress), close (become closed), collect (gather or collect), cook (transform and make suitable for consumption by heating), cut (separate with or as if with an instrument), decant (pour out), dilute (lessen the strength or flavor of a solution or mixture), drag (pull, as against a resistance), draw (bring, take, or pull out of a container or from under a cover), drink (take in liquids), drive (operate or control a vehicle), eat (eat a meal; take a meal), eat (take in solid food), edit (cut and assemble the components of), embrace (squeeze (someone) tightly in your arms, usually with fondness), exit (move out of or depart from), feed (give food to), fly (operate an airplane), function (perform as expected when applied), get (succeed in catching or seizing, especially after a chase), hit (hit against; come into sudden contact with), hit (cause to move by striking), hitch (to hook or entangle), hold (be the physical support of; carry the weight of), hold (have or hold in one's hands or grip), hop on (get up on the back of), ignite (cause to start burning; subject to fire or great heat), inject (give an injection to), inspect (look over carefully), install (set up for use), jab (poke or thrust abruptly), jump (bypass), kick (drive or propel with the foot), kick (thrash about or strike out with the feet), let go of (release, as from one's grip), lie (be located or situated somewhere; occupy a certain position), lie (be lying, be prostrate; be in a horizontal position), lie down (assume a reclining position), lift (take hold of something and move it to a different location), load (fill or place a load on), melt (reduce or cause to be reduced from a solid to a liquid state, usually by heating), move (cause to move or shift into a new position or place, both in a concrete and in an abstract sense), obstruct (block passage through), occupy (occupy the whole of), open (cause to open or to become open), open (start to operate or function or cause to start operating or functioning), operate (handle and cause

to function), pack (arrange in a container), paint (apply paint to; coat with paint), pare (remove the edges from and cut down to the desired size), photograph (record on photographic film), pick (look for and gather), plow (to break and turn over earth especially with a plow), produce (create or manufacture a man-made product), pull (apply force so as to cause motion towards the source of the motion), push (move with force, "He pushed the table into a corner"), push (press against forcefully without moving), put (put into a certain place or abstract location), race (compete in a race), raise (raise from a lower to a higher position), reach (reach a destination, either real or abstract), read (interpret something that is written or printed), rend (tear or be torn violently), repair (restore by replacing a part or putting together what is torn or broken), ride (sit and travel on the back of animal, usually while controlling its motions), ride (be carried or travel on or in a vehicle), score (gain points in a game), shoot (hit with a missile from a weapon), shoot (send forth suddenly, intensely, swiftly), sign (mark with one's signature; write one's name (on)), sit (be around, often idly or without specific purpose), sit (be seated), slide (move smoothly along a surface), smoke (inhale and exhale smoke from cigarettes, cigars, pipes), spin (revolve quickly and repeatedly around one's own axis), squeeze (press firmly), stand (hold one's ground; maintain a position; be steadfast or upright), stand (be standing; be upright), stand (remain inactive or immobile), steer (direct the course; determine the direction of travelling), sweep (move with sweeping, effortless, gliding motions), swerve (turn sharply; change direction abruptly), swing (make a big sweeping gesture or movement), switch off (cause to stop operating by disengaging a switch), throw (propel through the air), tie (fasten or secure with a rope, string, or cord), toast (propose a toast to), touch (perceive via the tactile sense), transport (move while supporting, either in a vehicle or in one's hands or on one's body), turn (let (something) fall or spill from a container), turn (cause to move around or rotate), turn (change orientation or direction, also in the abstract sense), unfold (spread out or open from a closed or folded state), walk (use one's feet to advance; advance by steps), wash (cleanse (one's body) with soap and water), wash (clean with some chemical process), work (shape, form, or improve a material), write (record data on a computer), write (mark or trace on a surface), adjust (make correspondent or conformable), flip (toss with a sharp movement so as to cause to turn over in the air), flush (rinse, clean, or empty with a liquid), grind (make a grating or grinding sound by rubbing together), groom (give a neat appearance to), herd (to gather and move a group of animals), hose (water with a hose), jump (cause to jump or leap), lasso (catch with a lasso), launch (to put (a boat or ship) on the water), lick (pass the tongue over), milk (take milk from female mammals), park (place temporarily), pay (give money, usually in exchange for goods or services), pet (stroke or caress gently), point (direct into a position for use), row (propel with oars), run (cause an animal to move fast), serve (put the ball into play), shear (shear the wool from), sip (drink in sips), smell (inhale the odor of; perceive by the olfactory sense), stick (pierce or penetrate or puncture with something pointed), stir (mix or add by stirring), straddle (sit or stand astride of), text on (send a message), type on (write by means of a keyboard with types), zip (close with a zipper), swim, talk on phone, work on computer, ski, surf, skateboard, dry (remove the moisture from and make dry), fly (travel through the air; be airborne), greet (express greetings upon meeting someone), hunt (pursue for food or sport (as of wild animals)), launch (propel with force), lose (fail to keep or to maintain; cease to have, either physically or in an abstract sense), prepare (educate for a future role or function), rub (scrape or rub as if to relieve itching), set up (get ready for a particular purpose or event), snog (touch with the lips or press the lips (against someone's mouth or other body part) as an expression of love, greeting, etc.), stop (stop from happening or developing), talk (exchange thoughts; talk with), teach (impart skills or knowledge to), walk (accompany or escort), watch (see or watch), watch (look attentively), wear (have on one's person), wield (handle effectively), tag (touch a player while he is holding the ball), wave (signal with the hands or nod).

