# OpenReview forum: "Beyond Object Recognition: A New Benchmark towards Object Concept Learning"
_ICLR.cc/2022/Conference — ICLR 2022 Submitted_

### Official Review · Reviewer_mgiJ · 2021-11-01

**Correctness:** 3
**Technical Novelty And Significance:** 2
**Empirical Novelty And Significance:** 2
**Recommendation:** 5
**Confidence:** 3

**Details Of Ethics Concerns:**

Potential biases in dataset should be addressed.

**Main Review:**

Overall I found that the writing of this paper can still be improved. For example, Table 1 is too small; there are too many bold texts in Section 3; Section 4.2 contains many paragraph subtitles, it will be better if they can be grouped according to whether they are applied to attribute prediction or affordance prediction, etc.

My primary concern about this paper, however, lies in its clarity in formulating the causal story. First, I would suggest the authors state clearly where do these causal graphs come from. Why should we predict attributes based on the category (figure 1)? Why should the image be generated based on the Category (figure 5)? Specifically,

- is the causal graph representing the nature? E.g., physics.
- is the causal graph representing the data generation/collection pipeline? E.g., we first sample a category, and then search for the images.
- is the causal graph representing human's reasoning of the properties? This is quite tricky. For example, the arrow could be attribute -> category (e.g., a yellow, oval citrus fruit with thick skin and fragrant, acidic juice => lemon), or affordance -> category (an object that can be used to hold or transport something. => container).

I guess the primary reason for introducing a specific Bayes net is for debiasing purposes. The key idea is to apply "interference" during inference to compensate the biases during training (e.g., category frequency, etc.)

However, I feel that this part of the paper is not well compared with other approaches towards "debiasing". For example,
https://openaccess.thecvf.com/content_CVPR_2020/papers/Wang_Towards_Fairness_in_Visual_Recognition_Effective_Strategies_for_Bias_Mitigation_CVPR_2020_paper.pdf
https://openaccess.thecvf.com/content/CVPR2021/papers/Ramaswamy_Fair_Attribute_Classification_Through_Latent_Space_De-Biasing_CVPR_2021_paper.pdf
These two papers discussed several methods for "remove biases." There are, of course, many other papers in this field, if the authors follow the pointers in these papers. I would like to see how the proposed model relate to these methods as well as how they compare empirically.

A minor thing about the metric: I am not sure how TDE/alpha-beta-TDE should be interpreted. why is this consistency score between network "weights" and "human labels" matter at all? This is especially tricky when all prediction networks take the image feature as their inputs.



**Summary Of The Paper:**

The primary contribution of this paper is two folds. First, the paper presents a new crowd-sourcing dataset that contains annotations for object categories, attributes, and affordances. Second, the paper presents a CNN training and inference pipeline that leverages "causal interference" to address potential biases.

**Summary Of The Review:**

I think the dataset part is a solid contribution of the paper. However, for the proposed model, I think the authors are missing important baselines and discussions with related work.

---

> ### Author Response · Authors · 2021-11-21
> **Response to Reviewer mgiJ (1/3)**
>
>
> > Overall I found that the writing of this paper can still be improved. For example, Table 1 is too small; there are too many bold texts in Section 3; Section 4.2 contains many paragraph subtitles, it will be better if they can be grouped according to whether they are applied to attribute prediction or affordance prediction, etc.
>
> **A**: Thanks, we have revised the paper including Tab. 1, Sec. 3, and 4.2.
>
> > My primary concern about this paper, however, lies in its clarity in formulating the causal story. First, I would suggest the authors state clearly where do these causal graphs come from. Why should we predict attributes based on the category (figure 1)?
> Why should the image be generated based on the Category (figure 5)? Specifically,
> > is the causal graph representing the nature? E.g., physics.
> > is the causal graph representing the data generation/collection pipeline? E.g., we first sample a category, and then search for the images.
> > is the causal graph representing human's reasoning of the properties?
> > This is quite tricky. For example, the arrow could be attribute -> category (e.g., a yellow, oval citrus fruit with thick skin and fragrant, acidic juice => lemon), or affordance -> category (an object that can be used to hold or transport something. => container).
>
>
> **A**: Thanks, we have clarified and added more discussion about our causal graph in the main text and supplementary.
> Our causal graph follows the human priors from our experts and crowdsourcing annotators. Some previous works also follow this before designing the method, such as [1].
> From the viewpoint of causal discovery [2][3][4][5], the above arcs (e.g., the inverted arc from attribute to category in the causal graph directed acyclic graph, DAG) are indeed possible. However, here, we mainly study the object concept learning problem, especially attribute and affordance learning. Thus, from the perspective of affordance learning, we think the arcs from category to attribute and affordance is more vital and meaningful to us.
>
>
>
> More specifically,
> (1) The causal graph follows the nature of **human-like recognition**. Given a new object, we refer to the category level commonsense first, then a specific instantiation [6]. Considering a simple case, when a human wants to find an apple to eat, he will pick up a fresh one instead of a bad one based on the recognition of object category and attribute.
> (2) Definitely, the causal graph reflects the **data collection pipeline**. We first select object categories before collecting images and complementing instances. After that, category-level *A* *B* and instance-level $\alpha \beta$ are annotated.
> (3) Different arcs are mainly attributed to **task context**, e.g., we have "image -> label" in image recognition task, but "label -> image" in image generation task.
> In OCL, the taxonomy viewpoint (how we define an object due to its attribute or affordance, e.g., a container is a container because it can be used to hold or transport something) lies out of the area of our concern. Instead, we consider more on the **physical** side of learning the property and functionality of an object for **intelligent agent/robot**. It will have a larger impact on boosting downstream tasks like robot manipulation.
>
>
>
>
> > I guess the primary reason for introducing a specific Bayes net is for debiasing purposes. The key idea is to apply "interference" during inference to compensate the biases during training (e.g., category frequency, etc.)
>
>
> **A**:
> The viewpoint is insightful! With the given causal graph, i.e., category to attribute/affordance, attribute to affordance, we can conduct the intervention smoothly during the inference and meanwhile debias the learning.
>
>
>
>
>
>
> **(To be continued)**

---

> > ### Author Response · Authors · 2021-11-21
> > **Response to Reviewer mgiJ (2/3)**
> >
> >
> > > However, I feel that this part of the paper is not well compared with other approaches towards "debiasing". For example, ... . These two papers discussed several methods for "remove biases." There are, of course, many other papers in this field, if the authors follow the pointers in these papers. I would like to see how the proposed model relates to these methods as well as how they compare empirically.
> > > (In Summary) For the proposed model, I think the authors are missing important baselines and discussions with related work.
> >
> >
> >
> > **A**: Thanks for the valuable suggestion!
> > Firstly, we respectfully reclaim the motivation of the OCRN is to follow the prior knowledge of the three levels of objects with a deep learning based causal graph model, to pursue the object understanding beyond the common direct mapping from pixels to labels.
> > To avoid the bias estimation such as in the **Simpson's paradox**[2]. Thus, we use intervention to deconfound the confounder *category* and exclusive the possible spurious bias and correlation imported bias from imbalanced object categories. Overall, we propose our OCRN in a causal inference perspective instead of the pure classification viewpoint, which also suits our causal graphical model well. Similar cases are also proposed in recent works like [3][4][5].
> > Moreover, the suggested methods from Reviewer XzUp and mgiJ are also able to address the bias alleviating.
> > For convenience, we also report the added experiment here.
> >
> > We regard $\alpha,\beta$ recognition as multiple independent binary classification tasks and implement some methods introduced in [7] on our strong baseline DM-V to reduce bias from object categories.
> > We use **mean bias amplification** (Amp.) in [8] as bias evaluation metric: small Amp means model suffers less from data category bias.
> > The test results are shown in the table below. The proposed OCRN has comparable or smaller bias amplification than the variants of DM-V since our model follows the causal graph and exploits the tools of causal inference, while most methods for category bias from the view of classification.
> >
> >
> > |       Model                        |                 Test Inference                   | $\alpha$ Amp. | $\beta$ Amp. |
> > |------------------------------------|--------------------------------------------------|:-------------:|:------------:|
> > | OCRN                               | $\text{argmax} _y P(y &#124; x)$                 |   **0.127**   |  **0.112**   |
> > | DM-V + Joint ND-way softmax        | $\text{argmax} _y \max _d P _{te}(y,d &#124; x)$ |     0.151     |    0.158     |
> > | DM-V + Joint ND-way softmax        | $\text{argmax} _y \sum _d P _{te}(y,d &#124; x)$ |     0.148     |    0.154     |
> > | DM-V + N-way classifier per domain | $\text{argmax} _y P _{te}(y &#124; d^*,x)$       |     0.135     |  **0.112**   |
> > | DM-V + N-way classifier per domain | $\text{argmax} _y \sum _d s (y,d,x)$             |     0.147     |    0.145     |
> >
> >
> > > A minor thing about the metric: I am not sure how TDE/alpha-beta-TDE should be interpreted. Why is this consistency score between network "weights" and "human labels" matter at all? This is especially tricky when all prediction networks take the image feature as their inputs.
> >
> >
> > **A**:
> > Thanks, we have re-written the related TDE metric section for better clarity.
> > TDE is to measure whether a model infers affordance with proper attention to the causality-related attribute. That said, when removing the attribute, the model is expected to have *large prediction difference further away from the ground truth*. This leads to the TDE score formulation as $max( P(\beta _ q) - P _ {TDE}^ {\alpha _ p}(\beta _ q), 0)$ when $GT _ {\beta q}=1$ while $max( P _ {TDE}^ {\alpha _ p}(\beta _ q) - P(\beta _ q), 0)$ when $GT _ {\beta q}=0$. Besides, we also propose a unified criterion to measure $ \alpha, \beta $ TDE three branches via multiplying the correctness of $\alpha,\beta$, and TDE score.
> > The "human labels" are the causality in the reasoning process of humans. The "weights" are a reflection of what causality our model has learned. We wish to connect two levels to achieve intelligence. Even all models take the same image feature, but they may learn different things under different inductive biases. However, humans always follow logic and causality. Even making mistakes, we seldom make the ones contrary to causality, but current machines do not.
> >
> >
> >
> > **(To be continued)**

---

> > > ### Author Response · Authors · 2021-11-21
> > > **Response to Reviewer mgiJ (3/3)**
> > >
> > >
> > >
> > > > (Ethics Concerns) Potential biases in dataset should be addressed.
> > >
> > > We have considered the bias issue in the construction of our dataset.
> > > 1. In our dataset, the existing datasets (ImageNet [9], COCO [10], aPY [11], SUN [12]) are open-sourced datasets and the images collected from Internet are publicly accessible too. The dataset is constructed for only non-commercial purposes. We will only provide URLs of these images to avoid copyright infringement.
> > > 2. During image collection, we choose images with general objects and are particularly careful with the images selection to avoid unsuitable content, private images, or implicit biases.
> > > 3. During annotation, the annotators cover different genders, ages, and fields of expertise to avoid potential annotation biases. And they are all informed how we will use the annotations in our research.
> > > The above discussions have been added to our appendix.
> > >
> > >
> > > ### References
> > >
> > > [1]: Yuke Zhu, Alireza Fathi, and Li Fei-Fei. Reasoning about object affordances in a knowledge base representation. In ECCV, 2014.
> > >
> > > [2]: Judea Pearl, Madelyn Glymour, and Nicholas P Jewell. Causal inference in statistics: A primer. John Wiley & Sons, 2016.
> > >
> > > [3]: Tan Wang, Jianqiang Huang, Hanwang Zhang, and Qianru Sun. Visual commonsense r-cnn. In CVPR, 2020a.
> > >
> > > [4]: Kaihua Tang, Jianqiang Huang, and Hanwang Zhang. Long-tailed classification by keeping the good and removing the bad momentum causal effect. arXiv preprint arXiv:2009.12991, 2020a.
> > >
> > > [5]: Kaihua Tang, Yulei Niu, Jianqiang Huang, Jiaxin Shi, and Hanwang Zhang. Unbiased scene graph generation from biased training. In CVPR, 2020b.
> > >
> > > [6]: Daniel N Osherson, Joshua Stern, Ormond Wilkie, Michael Stob, and Edward E Smith. Default probability. Cognitive Science, 15(2):251–269, 1991.
> > >
> > > [7]: Zeyu Wang, Klint Qinami, Ioannis Christos Karakozis, Kyle Genova, Prem Nair, Kenji Hata, and Olga Russakovsky. Towards fairness in visual recognition: Effective strategies for bias mitigation. In CVPR, 2020b.
> > >
> > > [8]: Jieyu Zhao, Tianlu Wang, Mark Yatskar, Vicente Ordonez, and Kai-Wei Chang. Men also like shopping: Reducing gender bias amplification using corpus-level constraints. arXiv preprint arXiv:1707.09457, 2017.
> > >
> > >
> > > [9]: Jia Deng, Wei Dong, Richard Socher, Li-Jia Li, Kai Li, and Li Fei-Fei. Imagenet: A large-scale hierarchical image database. In CVPR, 2009.
> > >
> > > [10]: Genevieve Patterson and James Hays. Coco attributes: Attributes for people, animals, and objects.
> > > In ECCV, 2016.
> > >
> > > [11]: Ali Farhadi, Ian Endres, Derek Hoiem, and David Forsyth. Describing objects by their attributes. In CVPR, 2009.
> > >
> > > [12]: Jianxiong Xiao, James Hays, Krista A Ehinger, Aude Oliva, and Antonio Torralba. Sun database: Large-scale scene recognition from abbey to zoo. In CVPR, 2010.

---

> ### Comment · Reviewer_mgiJ · 2021-11-28
> **Re: Author Response**
>
> I have carefully read the authors' responses. I think the authors have made a good justification for their model design (especially the causal graph part), and I do value the additional experiments on the debiasing experiments. However, I don't think this paper is in a good shape for ICLR publication.
>
> Specifically, I am not convinced by the authors' claim on human recognition: the cited paper [6] by Osherson et al. is not a relevant paper for visual recognition. The second point about "causal graph reflects data collection process" is also potentially dangerous: should I view the causal graph approach as a way to tell the model how I have collected the data?
>
> In general, I think the causal relationship between "category" and "attribute" isn't obvious and broadly applicable. For the affordance labels in the created dataset, the causal arcs from category and attribute to affordance are better justified (On a side note, I agree with the authors that maybe concepts such as "container" should be excluded from their scope---I hope authors will include discussions about this in the paper)
>
> However, the debiasing experiments are not particularly supportive for the affordance learning: the model does not significantly outperform baselines in this task: $\beta$ Amp. Interestingly, the model shows significant performance in terms of attribute classification debiasing.
>
> In summary, I do value the contribution of creating a large-scale dataset for object attributes and affordance. However, the model presentation still lacks a clear justification for the usage of causal graphs. The added experiments about debiasing baselines added a lot to the paper, but I think the paper needs a major revision to incorporate more discussions about related works in this domain: for example, mathematically, are the domain independent classifiers (e.g., N-way per domain) an approximation for the "causal intervention"? Meanwhile, if the authors are going to compare their method against other debiasing methods, it should also include evaluations on other existing datasets.
>
> Overall, I appreciate the authors for clarifications and the changes. However, I believe there are still many things to be addressed and this paper needs a major revision and resubmission for another round of review. I decide to retain my score.

---

> > ### Author Response · Authors · 2021-11-30
> > **Response to Reviewer**
> >
> > Thanks for the feedback!
> >
> > We are pleased that our additional experiments have addressed some concerns about debiasing from two reviewers.
> > Though our main contribution is the causal benchmark on real-world object data, and the proposed baseline OCRN is designed for the causal inference of object concept instead of the attribute/affordance imbalanced classification and debiasing, we believe these results and more discussion suggested by the reviewers would make our method part more comprehensive.
> > Thanks again for this valuable suggestion. In our final version, we will add more discussion about debiasing literature in the related work and the experiment parts.
> >
> > In terms of the causal graph prior, we will add more discussion about the different causal structures in different scenarios, and analyze possible different method designs according to the different causal structures. We cited Osherson et al. [6] to give an example that humans can understand the world in a process similar to our prior causal relations. As for the potentially dangerous, we give the data collection as an intuitive example, in practice, the annotation process is much more complex and basically impossible to hack (e.g., voting, rechecking, many times relabeling, hundreds of categories, attributes, and affordances and their relations). This can also be verified by the poor performances of the baselines. But we will clarify these in our final version to avoid possible ambiguities.
> >
> > Best,
> >
> > Authors

---

### Official Review · Reviewer_mY2o · 2021-11-03

**Correctness:** 3
**Technical Novelty And Significance:** 4
**Empirical Novelty And Significance:** 3
**Recommendation:** 6
**Confidence:** 3

**Main Review:**

Strengths:
1. Having such a large-scale dataset with detailed annotations and causal relationships of visual concepts (attributes and affordances in this paper) is valuable for the entire community. It will advance future research on causal learning from visual data.

2. Data and code will be publicly available (they are available for review).

Weaknesses:

First of all, I have to admit I have limited knowledge about causal learning. I found some technical details are missing, particularly in Section 4 and 5, which make the technical part not easy to follow. Moreover, it would be great if more information could be provided in the paper (or in the supplementary material). Detailed comments can be found below.

1. For the data annotation, more annotators were hired if the first set of annotators could not reach an agreement. What is the percentage of such confusing cases?

2. What is the total cost of building such a dataset? This is useful as a reference if people would like to build a similar dataset in the future.

3. Although experts were involved in designing the data annotation process, for example, defining the object states, it is not clear whether there is any way to formally monitor the quality of the annotations.

4. Object states are used in the data annotation process for the instance-level affordance and causal relation parts. I would love to see the annotations of object states for such a benchmark dataset to be released as well. But they are intentionally excluded in the benchmark dataset and being used in the baseline model. Here are two questions:
    a) Since object states were used in the annotation process, is it possible that without using the states, some annotations of afforances and causal relations may be ambiguous or do not hold at all? For example, I may argue that both the apples in the middle column of Fig 4 can not be *directly* eaten as they need to be washed first.
    b) According to the experimental results, the performance of the proposed baseline is still far from satisfactory. What if we explicity consider object states as part of the causal structure? Will it bring any improvement?


5. What are the context in Fig. 4? Are they attributes or object states?

6. For the task definition in Eq.(1), it is not clear about what the concrete tasks are. It would be helpful to explicitly explain/define them.

7. In Eq.(7) and Fig. 6, $f_{\alpha}'$ is introduced but without formal explanations. Particularly in Fig. 6, what are $f_{\alpha_1}$, $f_{\alpha_2}$, and $f_{\alpha_p}$? How are they different from $f_{\alpha}$? Where is the binary classifier $g_{\alpha_p}$? Is its loss part of $L_{\alpha}$?

8. Clarifications are needed for the definition of TDE. From my understanding, when $GT_{\beta_q}=1$, it means the affordance $\beta_q$ exists in the image. As a result, a high score is desired from the model's predicted score. Ideally, by explicitly considering the causal relationship of $\alpha_p$ and $\beta_q$, $P_{TDE}^{\alpha_p}(\beta_q)$ should be higher than $P(\beta_q)$. So TDE should be defined as $max(P_{TDE}^{\alpha_p}(\beta_q) - P(\beta_q), 0)$. Similarly, for the case where $GT_{\beta_q}=0$, it should be $max(P(\beta_q) - P_{TDE}^{\alpha_p}(\beta_q), 0)$.

**Summary Of The Paper:**

This paper proposes a new large-scale benchmark dataset for object concept learning, which consists of recognizing attributes, affordances, and their causal raltions about objects in input images. Detailed annotations of object categories, attributes and affordances on both category and instance levels, and their causal relations (on the instance level) are provided. This new dataset will be helpful for the community to advance the research of causal learning from visual data.

A strong baseline method is also proposed that explicitly considers the causal structure and concept instantiation of object categories, attributes, and affordances. Although it shows better results than other baselines, there is still great room for future improvement.

**Summary Of The Review:**

The issues mentioned in the weaknesses part above are minor. They do not affect the value of the proposed benchmark dataset and baseline algorithm. But it would be great if they could be addressed.

---

> ### Author Response · Authors · 2021-11-21
> **Response to Reviewer mY2o (1/2)**
>
>
> > I found some technical details are missing, particularly in Section 4 and 5, which make the technical part not easy to follow.
>
> **A**: Thanks! We have revised our paper, especially Sec. 4 and 5. Please refer to the pdf for more details.
>
> > What is the total cost of building such a dataset? This is useful as a reference if people would like to build a similar dataset in the future.
> > For the data annotation, more annotators were hired if the first set of annotators could not reach an agreement. What is the percentage of such confusing cases?
>
>
> **A**:
> Thank you for your interest and we are glad to show more statistics in dataset construction. We mainly introduce $A, B, \alpha, \beta$, causality annotation, which are divided into multiple finer-grained small sets in our pipeline. Generally, we have 13, 19, 124, 140, 85 annotation sets (381 totally) for $A, B, \alpha, \beta$, causality annotation respectively. We assign each small set to 2 annotators. However, considering the controversial situations introduced, part of the annotation are confused cases based on their results. In the whole process, 9.6\% of $A$, 7.7\% of $B$, 5.2\% of $\alpha$, 7.9\% of $\beta$, and 13.7\% of causality are confusing and re-assigned to additional annotators.
>
>
> In total, there are 89 part-time annotators involved. Each stage of the annotation (1. images and boxes; 2. $A$ and $B$; 3. object states; 4. $\alpha$ and $\beta$) takes 1-2 months for crowdsourcing and checking.
>
>
> > Although experts were involved in designing the data annotation process, for example, defining the object states, it is not clear whether there is any way to formally monitor the quality of the annotations.
>
> **A**: The expert-involved annotations are labeled and checked by multiple experts. If the experts give different annotations, it will be checked by more other experts until consensus. During crowdsourcing, the annotators can give feedbacks to the experts' annotation and mend the errors.
>
>
> > I would love to see the annotations of object states for such a benchmark dataset to be released as well.
>
> **A**: Thanks! We have provided the full list of object states in the *Appendix*, and we will release the object state annotation too.
>
> > But they (object states) are intentionally excluded in the benchmark dataset and being used in the baseline model.
> > According to the experimental results, the performance of the proposed baseline is still far from satisfactory. What if we explicitly consider object states as part of the causal structure? Will it bring any improvement?
>
>
> **A**: We did not use object states in our model because there is a *compositional zero-shot* problem for object-state pairs, i.e., there can be *unseen* states in real-world data. Differently, affordances are more general. The models explicitly incorporating object states will fail to generalize to these zero-shot states and it adds to the object category bias.
> In experiments, the state supervision during training would indeed slightly improve the affordance recognition performance, since instances in the *same state* lies in a tight cluster in affordance label space. But this will hurt the TDE performance greatly.
>
>
> > Since object states were used in the annotation process, is it possible that without using the states, some annotations of affordances and causal relations may be ambiguous or do not hold at all? For example, I may argue that both the apples in the middle column of Fig. 4 can not be directly eaten as they need to be washed first.
>
> **A**: Thanks! It's possible that the original state proposition can't cover all instances in the category. Thus we allow annotators to dynamically expand the state list for the finer-grained label. And we are reminded to recheck each instance with its state and $\beta$ label to avoid ambiguity and improve quality.
> Causality annotation is a more difficult procedure but we employ more annotations and put more effort. We attempt to mine clearer and stronger causal relations and only give a positive causal label when annotators reach a consensus. Inevitably, there is controversy or always some corner cases in our annotation like your argument of Fig. 4 (Currently Fig.3), but we assign positive labels with the agreement of the majority. Statistically, we can see the final causal relation labels stand for the common understanding of more humans with a high probability.
>
>
>
> **(To be continued)**

---

> > ### Author Response · Authors · 2021-11-21
> > **Response to Reviewer mY2o (2/2)**
> >
> >
> > > What are the context in Fig. 4? Are they attributes or object states?
> >
> > **A**: Attributes and object states are both contained in the context of Fig. 4. We have revised the context in Fig. 4 to make it clear (Fig.3 in the revised version).
> >
> >
> > > For the task definition in Eq.(1), it is not clear about what the concrete tasks are. It would be helpful to explicitly explain/define them.
> >
> >
> > **A**: Thanks, we have revised the formulation in Sec. 4.1 to better illustrate our problem and task, including the attribute/affordance recognition and TDE-based causal relation evaluation. Please refer to the pdf for more details.
> >
> > > In Eq. (7) and Fig. 6, $f _{\alpha}'$ is introduced but without formal explanations. Particularly in Fig. 6, what are $f _{\alpha _1}$, $f _{\alpha _2}$, and $f _{\alpha _p}$? How are they different from $f _{\alpha}$? Where is the binary classifier $g _{\alpha _p}$? Is its loss part of $L _{\alpha}$?
> >
> >
> > **A**: Thanks, we will clarify the $f _{\alpha}'$ in Sec. 4.2.
> > $f _{\alpha}'$ is the aggregation of features $f _{\alpha _p}$ and $f _{\alpha _p}$ is the feature of $p^th$ attribute. In this way, we can **control** a specific attribute and observe its total direct effect on affordance.
> >
> > The binary classifiers $g _{\alpha _p}$ are applied on $f _{\alpha _p}$ to predict the $p^{th}$ attribute and their losses are included in $L _{\alpha}$.
> >
> >
> >
> >
> > > Clarifications are needed for the definition of TDE. From my understanding, when $GT_{\beta _q}=1$, it means the affordance $\beta _q$ exists in the image. As a result, a high score is desired from the model's predicted score. Ideally, by explicitly considering the causal relationship of $\alpha _p$ and $\beta _q$, $P _{TDE} ^ {\alpha _ p} (\beta _ q)$ should be higher than $P(\beta _ q)$. So TDE should be defined as $max(P _ {TDE} ^ {\alpha _ p}(\beta _ q) - P(\beta _ q), 0)$. Similarly, for the case where $GT _ {\beta _ q}=0$, it should be $max(P(\beta _ q) - P _ {TDE}^ {\alpha _ p}(\beta _ q), 0)$.
> >
> >
> > **A**: We have revised this part for better clarification. Here, we explain the TDE metric as follows.
> >
> > For an attribute $\alpha _ p$ and affordance $\beta _ q$, if they have a strong GT causal relation. That said, $\alpha _ p$ would affect $\beta _ q$ a lot. For example, for a *cup*, attribute *broken* has strong negative effect on affordance *holding water*. For an intelligent agent, if it learns the causality well, when $\alpha _ p$ changes, it should be able to infer the change of $\beta _ q$ well too. For instance, when broken is False, the cup can hold water (True); if not, $\beta _ q$ also should change from True to False.
> > In the above case, we should see a larger $\beta _ q$ **score change** when **removing** the effect of the $\alpha _ p$. Thus, we propose to use TDE to measure this kind of affordance score change when attribute feature changes.
> > Given a detailed case, for an object, $\alpha _ p$ and $\beta _ q$ have a strong causal relation, positive or negative influence (*broken* is negative to *holding* for *cup*).
> >
> > **First**, a model will predict $P(\beta _ q)$ given the features of all kinds of attributes (including $\alpha _ p$). The metric of $\beta$ would evaluate whether it can predict $P(\beta _ q)$ well.
> > **Second**, if we exclude the feature (information) of $\alpha _ p$ in model input, the model should think the prediction $P(\beta _ q)$ should change to its opposite. In detail, if $GT _ {\beta q}=1$, $P(\beta _ q)$ is high (e.g., 0.8, model performs good), after removing $\alpha _ p$ feature (**counterfactual** scenario), the changed score $P _ {TDE}^ {\alpha _ p}(\beta _ q)$ should be smaller than 0.8. Thus, we use $max(P(\beta _ q) - P _ {TDE}^ {\alpha _ p}(\beta _ q), 0)$ when $GT _ {\beta q} = 1$ as the TDE score. That is, the larger of $P(\beta _ q) - P _ {TDE}^ {\alpha _ p}(\beta _ q)$ (>0), the better.
> > **Third**, if $GT _ {\beta q}=0$, the initial prediction $P(\beta _ q)$ should be small (e.g., 0.1). After removing $\alpha _ p$ feature, the changed score $P _ {TDE}^{\alpha _p}(\beta _q)$ should be larger than 0.1. Thus, here we use $max( P _ {TDE}^ {\alpha _ p}(\beta _ q) - P(\beta _ q), 0)$ as the TDE score. The larger of $P _ {TDE}^ {\alpha _ p}(\beta _ q) - P(\beta _ q)$ (>0), the better.

---

### Official Review · Reviewer_xQXf · 2021-11-03

**Correctness:** 3
**Technical Novelty And Significance:** 2
**Empirical Novelty And Significance:** 2
**Recommendation:** 3
**Confidence:** 4

**Main Review:**

The strengths of the paper are as follows:

- (Very) Important problem.
- A new dataset is always a good contribution. Specifically, the work appears to have done the extra mile in combining lots of different and different data sources.

The overall weaknesses of the paper are as follows:
- The technical contribution is low. Basically, this is a dataset/task proposition paper. This is also the authors' summary.
- In that sense, it feels that the paper is not a direct fit to ICLR, which focuses more on technical contribution.
- While it is claimed that the work is about causality, the causality in this context is rather learned with standard correlation-based approaches, which minimize standard learning objectives.
- In eq 3, the notation is not entirely clear to me, or I am misunderstanding something. If A stands for category-level attributes, their number is different than the number of object categories. How can then be that $P(A_j|O_i)=1$ $iff i=j$, since these two indices run over different sets?
- It is not entirely clear what is the baseline supposed to do? Basically, learn 'causal mechanisms' such that to attain 'perfect' recognition of the category-level, instance level attributes and affordances, and category label? And to do so, the do-operation intervention runs over all attributes/affordances in (3) and (6)? What exactly is the motivation of this intervention, as in selecting one attribute/affordance over another?
- The writing is not always very clear. For instance, I am not sure what is the x axis of figure 8 and 9, I could not find it in the caption? Is it the number of training data used to train the model, and if yes, why is nearly zero for a long period in the figure?

**Summary Of The Paper:**

In short, the paper proposes a new task, that of object concept learning. Object concept learning is a basically a combination of object classification, attribute classification of said object, and affordance classification of said object, both at a category (affordance of any cup) and an instance (affordance of this cup) level. To this end, the paper proposes a dataset, which starts from existing ones and extends them accordingly. Further, it proposes a baseline method that is somewhat inspired by do-calculus (Pearl). Experiments show positive trends.

**Summary Of The Review:**

In summary, I think the paper brings little technical innovation, and as such it should not be accepted.

---

> ### Author Response · Authors · 2021-11-21
> **Response to Reviewer xQXf (1/2)**
>
>
> > The technical contribution is low. Basically, this is a dataset/task proposition paper. This is also the authors' summary. In that sense, it feels that the paper is not a direct fit to ICLR, which focuses more on technical contribution.
>
> **A**: We respectfully disagree with your comment. We believe ICLR is an open and diversifying community and welcome a wide range of subject areas including dataset and benchmark. Besides, there are many previous published dataset and benchmark papers, to name just a few: [1][2][3].
>
>
>
> > While it is claimed that the work is about causality, the causality in this context is rather learned with standard correlation-based approaches, which minimize standard learning objectives.
>
>
> **A**:
> We respectfully explain that, for attribute/affordance recognition only, all methods indeed adopt the labels to learn knowledge from the data. Moreover, in the evaluation of causal relation, only the "w/ $L_{TDE}$" adopts the causal relation labels. We hope the models can self-learn to mine and follow the intrinsic causalities. Thus, we design the TDE operation to evaluate this ability. Many works like [4][5][6] also try to marry supervised deep learning and causal inference which are similar to our OCRN method.
> Besides, current techniques especially the deep learning methods perform unsatisfying in our OCL task, and all show poor ability to capture the causality. We think our main aim and contribution here is to reveal this important problem via our OCL dataset instead of directly addressing the task we propose. A lot of works should be done to prompt this field which we hope our OCL can support.
>
>
>
>
> > In eq 3, the notation is not entirely clear to me, or I am misunderstanding something. If $A$ stands for category-level attributes, their number is different than the number of object categories. How can then be that $P(A_j|O_i)=1$ iff $i=j$ since these two indices run over different sets?
>
> **A**: In this equation, $O _i$ stands for $i^{th}$ category, and $A _j$ stands for the category-attribute vector of $j^{th}$ category, so the indices run over the same set. We will clarify this in Sec. 4.2.
>
>
> > It is not entirely clear what is the baseline supposed to do?
> > Basically, learn 'causal mechanisms' such that to attain 'perfect' recognition of the category-level, instance-level attributes and affordances, and category label?
> > What exactly is the motivation of this intervention, as in selecting one attribute/affordance over another?
>
> **A**: Baselines are supposed to give an accurate prediction on $\alpha, \beta$ and *simultaneously* obey the causal relations between $\alpha$ and $\beta$ in its reasoning procedure additionally. Category-level attributes and affordances are not evaluated for all models but as an intermediate for some methods.
> *There is only one right way, but many wrong ways*. Perfect $\alpha,\beta$ recognition, and the causal mechanism is only an ideal expectation, but far beyond achieved. Humans can reason out the answers, even sometimes make mistakes, still, stick with logic and causality. But current machines cannot ensure this. Thus, the task deserves more attention as we expect the model to respect causality in recognition but not to simply fit the dataset.
>
> > And to do so, the do-operation intervention runs over all attributes/affordances in (3) and (6)?
>
> **A**:
> We have updated the text to avoid ambiguity.
> Do-operator is operated over the object category, not the attribute class. It performs in deconfounding object category to achieve generalized $\alpha, \beta$ recognition.
>
>
>
>
> **(To be continued)**

---

> > ### Author Response · Authors · 2021-11-21
> > **Response to Reviewer xQXf (2/2)**
> >
> >
> > > The writing is not always very clear. For instance, I am not sure what is the x axis of figure 8 and 9, I could not find it in the caption?
> > Is it the number of training data used to train the model, and if yes, why is nearly zero for a long period in the figure?
> >
> > **A**: Thanks for your comment, we have added more explanations of the two figures.
> > The x-axes are attribute classes or affordance classes and the y-axes are bias indicator $b(O,\alpha)$ introduced by [7].
> > Take the current Fig. 7 (Fig. 8 in our original submission) as an example, on category $O=$*frying pan*, for each attribute class $\alpha$, we compute the bias of $\alpha$ towards $O$ of the dataset (green curve) or our learned model (blue or orange curves). As discussed in [7], the closer model bias and data bias are, the more biased the model is.
> > In the figure, the orange curve (model w/o deconfounding) is closer than the blue curve (model w/ deconfounding) to the green curve, so the deconfouding of our OCRN alleviates the bias learned by the model.
> >
> >
> >
> > ### References
> >
> > [1] GLUE: A Multi-Task Benchmark and Analysis Platform for Natural Language Understanding. ICLR 2019.
> >
> > [2] Benchmarking neural network robustness to common corruptions and perturbations. ICLR 2019.
> >
> > [3] IsarStep: a Benchmark for High-level Mathematical Reasoning. ICLR 2021.
> >
> > [4]: Tan Wang, Jianqiang Huang, Hanwang Zhang, and Qianru Sun. Visual commonsense r-cnn. In CVPR, 2020a.
> >
> > [5]: Kaihua Tang, Jianqiang Huang, and Hanwang Zhang. Long-tailed classification by keeping the good and removing the bad momentum causal effect. arXiv preprint arXiv:2009.12991, 2020a.
> >
> > [6]: Kaihua Tang, Yulei Niu, Jianqiang Huang, Jiaxin Shi, and Hanwang Zhang. Unbiased scene graph generation from biased training. In CVPR, 2020b.
> >
> > [7]: Jieyu Zhao, Tianlu Wang, Mark Yatskar, Vicente Ordonez, and Kai-Wei Chang. Men also like shopping: Reducing gender bias amplification using corpus-level constraints. arXiv preprint arXiv:1707.09457, 2017.

---

### Official Review · Reviewer_XzUp · 2021-11-04

**Correctness:** 3
**Technical Novelty And Significance:** 3
**Empirical Novelty And Significance:** 3
**Recommendation:** 6
**Confidence:** 4

**Main Review:**

+ves:
+ Present-day ML models lay a great emphasis on categorizing objects based solely on appearance. In case of distribution shift in a real-world dataset, such models find it difficult to generalize. A deeper understanding of objects based on their physical properties and affordances can avoid such problems, and the dataset introduced in this paper is a notable step in the right direction.
+ The constructed dataset is large-scale which will encourage the generalization of models towards real-world data and large-scale applications.

Concerns:
- The paper has adopted a causal intervention method to solve for category biasing. However, there are simpler methods to tackle such problems (undersampling/oversampling majority/minority classes, generating synthetic samples etc.). How is the proposed method to these approaches? Why is a causal approach necessary? For a dataset that may be of wide use, it may also be good to not make its description more complex than it should be.
- Surprisingly, for an image dataset paper, there were very few examples of the images in the paper. It may have been nice to see a few sample images in the supplementary material along with a clear description of their annotations.
- The paper presentation is very dense. While I appreciate the detailed description of every aspect, it at times seems one level of detail beyond what is required. Besides, for what seems like a dataset created with careful thought, it would have been good to see more examples from the dataset, or a running example to explain the decisions made for the dataset design.
- Similarly, in Sections 4 and 5, the many notations and inline equations make it difficult to follow. It may be a good idea to write problem formulation (or include them in the task overview), which lays down the assumptions, notations and objectives clearly and concisely, and then discuss the rest of the work.
- The notion of causal relations in real-world scenes can be tricky and complex (for e.g, the notion of context -- water can be causal for a swimming fish making context causal). In the few examples presented in the paper, it was not very clear whether the relationships between the attributes and the affordances in the image are really causal. The paper mentions that two annotators were used to label such causal relationships -- I am not sure if this may be sufficient.

Minor comments:
* Section 4.1 the sentence, “The ɑ, 𝛃 probabilities ...” seems incomplete.
* Section 4.2 “Attribute Estimation Debiasing” there is inconsistency in the notation of f_Oi.

**Summary Of The Paper:**

This paper introduces a large annotated dataset for object concept learning. The proposed dataset contains annotation at two levels of granularity (category and instance level). The dataset also provides causal relations between object attributes and their affordances. The paper also provides a thorough insight into their annotation process, and introduces a baseline method called the Object Concept Reasoning Network (OCRN) based on causal intervention and instantiation. The proposed dataset is large-scale for similar ones in this space, and can be of great help to researchers working in the area of concept learning and compositionality.

**Summary Of The Review:**

In general, the paper presents a dataset, which is a positive step towards concept learning tasks and will hopefully encourage future works to move beyond object categorization based only on appearance. The paper presentation seems too dense, and it’d be good to see some concrete directions of presenting the paper better to make it more accepted by the community.

I will wait to hear back from the authors and other reviewers, and accordingly adjust my score.

---

> ### Author Response · Authors · 2021-11-21
> **Response to Reviewer XzUp (1/2)**
>
>
> > The paper has adopted a causal intervention method to solve for category biasing. However, there are simpler methods to tackle such problems (undersampling/oversampling majority/minority classes, generating synthetic samples, etc.).
> > How is the proposed method to these approaches?
> > Why is a causal approach necessary? For a dataset that may be of wide use, it may also be good to not make its description more complex than it should be.
>
> **A**: Thanks for the valuable suggestion!
> Firstly, we respectfully explain the motivation of the OCRN is to follow the prior knowledge of the three levels of objects with a deep learning based causal graph model, to pursue the object understanding beyond the common direct mapping from pixels to labels. To avoid the bias estimation such as in the **Simpson's paradox** [1]. Thus, we use intervention to deconfound the confounder *category* and exclusive the possible spurious bias and correlation imported bias from imbalanced object categories. Overall, we propose our OCRN in a causal inference perspective instead of the pure classification viewpoint, which also suits our causal graphical model well. Similar cases are also proposed in recent works like [2][3][4].
> Moreover, the suggested methods from Reviewer XzUp and mgiJ are also able to address the bias alleviating. Therefore, we further add to conduct the experiments as follows.
>
>
> We regard $\alpha,\beta$ recognition as multiple independent binary classification tasks and implement some methods introduced in [5] on our strong baseline DM-V to reduce bias from object categories.
> We use **mean bias amplification** (Amp.) in [6] as the bias evaluation metric: small Amp means model suffers less from data category bias.
> The test results are shown in the table below. The propose OCRN has comparable or smaller bias amplification than the variants of DM-V, since our model follows the causal graph and exploits the tools of causal inference, while most methods for category bias from the view of classification.
>
>
>
> |       Model                        |                 Test Inference                   | $\alpha$ Amp. | $\beta$ Amp. |
> |------------------------------------|--------------------------------------------------|:-------------:|:------------:|
> | OCRN                               | $\text{argmax} _y P(y &#124; x)$                 |   **0.127**   |  **0.112**   |
> | DM-V + Joint ND-way softmax        | $\text{argmax} _y \max _d P _{te}(y,d &#124; x)$ |     0.151     |    0.158     |
> | DM-V + Joint ND-way softmax        | $\text{argmax} _y \sum _d P _{te}(y,d &#124; x)$ |     0.148     |    0.154     |
> | DM-V + N-way classifier per domain | $\text{argmax} _y P _{te}(y &#124; d^*,x)$       |     0.135     |  **0.112**   |
> | DM-V + N-way classifier per domain | $\text{argmax} _y \sum _d s (y,d,x)$             |     0.147     |    0.145     |
>
>
> > The paper presentation is very dense. While I appreciate the detailed description of every aspect, it at times seems one level of detail beyond what is required.
>
> **A**: Thanks, we have revised Sec. 3 to make it more concise and have moved some details to supplementary.
>
>
>
> > Surprisingly, for an image dataset paper, there were very few examples of the images in the paper. It may have been nice to see a few sample images in the supplementary material along with a clear description of their annotations.
> > Besides, for what seems like a dataset created with careful thought, it would have been good to see more examples from the dataset, or a running example to explain the decisions made for the dataset design.
>
> **A**: We have added more examples of category, $\alpha$, $\beta$, object state, causal rule in Sec. A.3.10. We also provide a running example of dataset construction in Sec. A.2
>
>
>
> **(To be continued)**

---

> > ### Author Response · Authors · 2021-11-21
> > **Response to Reviewer XzUp (2/2)**
> >
> >
> > > Similarly, in Sections 4 and 5, the many notations and inline equations make it difficult to follow. It may be a good idea to write problem formulation (or include them in the task overview), which lays down the assumptions, notations, and objectives clearly and concisely, and then discuss the rest of the work.
> > > The paper presentation seems too dense, and it’d be good to see some concrete directions of presenting the paper better to make it more accepted by the community.
> >
> > **A**: Thanks, we have revised Sec. 4 to add a clearer formulation of our OCL task and the evaluation of recognition and causal inference both. Besides, Sec. 3, 4, and 5 are all revised to make the illustration more concise and easy to follow. Please refer to the PDF for more details.
> >
> >
> >
> >
> > > The notion of causal relations in real-world scenes can be tricky and complex (for e.g, the notion of context -- water can be causal for a swimming fish making context causal). In the few examples presented in the paper, it was not very clear whether the relationships between the attributes and the affordances in the image are causal.
> > > The paper mentions that two annotators were used to label such causal relationships -- I am not sure if this may be sufficient.
> >
> > **A** : Definitely, we agree that annotating causality in the real world is extremely difficult. In data annotation, we have met numerous ambiguities and difficulties to confirm the "right" causal relations. To address these challenges, we follow the following principles:
> > (1) Firstly, we only emphasize on **clear** and **strong** causal relations via crowdsourcing but omit the vague ones.
> > (2) Second, we take an object **affordance-centric** viewpoint to look at the possible causal relations.
> > (3) We would rather discard than condone the controversial situation in the annotation.
> > (4) We only focus on the simple relations between **one** attribute and **one** affordance, instead of the very complex compositions of multiple attributes and affordances which are almost impossible to annotate.
> > Therefore, we finally find that we can label **a very small percentage** of all edges with the whole causal graph consisting of so many nodes (category, attributes, affordances, contexts, etc.) while keeping the quality.
> > Besides, we have added more cases of the causal relation labels in the supplementary to give more examples.
> >
> > As for the two annotators for each case, we explain that in our annotation process here.
> >
> > We divide $A, B, \alpha, \beta,$ causality annotation into multiple finer-grained small sets in our pipeline. Generally, we have 13, 19, 124, 140, 85 annotation sets (381 totally) for $A,B,\alpha,\beta$, causality annotation respectively. We assign each small set to 2 annotators. However, considering the controversial situations introduced, part of the annotation are confused cases based on their results. In the whole process, 9.6\% of $A$, 7.7\% of $B$, 5.2\% of $\alpha$, 7.9\% of $\beta$, and 13.7\% of causality are confusing and re-assigned to additional annotators. These indeterminable ones will be sent to two extra two annotators until making an agreement. The quality of the dataset is guaranteed by a low confusion ratio and multiple refining stages.
> >
> >
> > > Section 4.1 the sentence, “The $\alpha, \beta$ probabilities ...” seems incomplete.
> > > Section 4.2 “Attribute Estimation Debiasing” there is inconsistency in the notation of $f_{O_i}$.
> >
> > **A**: Thanks, we have revised these two sentences. Please see the blue words.
> >
> > ### References
> >
> > [1]: Judea Pearl, Madelyn Glymour, and Nicholas P Jewell. Causal inference in statistics: A primer. John Wiley & Sons, 2016.
> >
> > [2]: Tan Wang, Jianqiang Huang, Hanwang Zhang, and Qianru Sun. Visual commonsense r-cnn. In CVPR, 2020a.
> >
> > [3]: Kaihua Tang, Jianqiang Huang, and Hanwang Zhang. Long-tailed classification by keeping the good and removing the bad momentum causal effect. arXiv preprint arXiv:2009.12991, 2020a.
> >
> > [4]: Kaihua Tang, Yulei Niu, Jianqiang Huang, Jiaxin Shi, and Hanwang Zhang. Unbiased scene graph generation from biased training. In CVPR, 2020b.
> >
> > [5]: Zeyu Wang, Klint Qinami, Ioannis Christos Karakozis, Kyle Genova, Prem Nair, Kenji Hata, and Olga Russakovsky. Towards fairness in visual recognition: Effective strategies for bias mitigation. In CVPR, 2020b.
> >
> > [6]: Jieyu Zhao, Tianlu Wang, Mark Yatskar, Vicente Ordonez, and Kai-Wei Chang. Men also like shopping: Reducing gender bias amplification using corpus-level constraints. arXiv preprint arXiv:1707.09457, 2017.

---

### Author Response · Authors · 2021-11-21
**General response and revised manuscript**


Thanks for all your insightful comments. We appreciate your time on our work.
We are encouraged they found the problem we study is very **important** (R-xQXf), our dataset is **large-scale** (R-XzUp, R-xQXf) and **can be great help** to concept learning, compositionality, and causal learning community (R-XzUp, R-mY2o), which is a **notable, valuable, and solid** step (R-XzUp, R-mY2o, R-mgiJ) towards **real-world data and large-scale applications** (R-XzUp), and our annotation process is **thoroughly described** (R-XzUp). Moreover, we are glad they found our analysis about object causal structure and concept instantiation is **explicit** and will **advance** future research on causal learning (R-mY2o).

Here, we first reclaim our main contributions: our main aim is to study how to benchmark causal reasoning on **real data**, especially for object understanding which is super vital for embodied AI. To provide cues for future study, besides building the datasets, we also propose a basic system consisting of many baselines and analyze them on OCL.

We have revised our paper according to the suggestions to improve the clarity and readability. Below are the major updates:

- (Sec.3) Some details of annotation are moved to *Appendix Sec. A.2*.
- (Sec.4) Add a clearer formulation of the OCL task and make the equations and illustration more concise.
- (Sec.5) Revise the introduction of TDE. Some details are presented in *Appendix Sec. A.5*.
- (Appendix Sec. A.3) Add more examples of the category, $\alpha$, $\beta$, and causal rules.
- (Appendix Sec. A.8) Discussion on debiasing.
- (Appendix Sec. A.9) Discussion on object states.
- (Appendix Sec. A.10) Discussion on causality and the causal graph.

Best regards,

Authors

---

### Decision · Program_Chairs · 2022-01-20

**Decision:**

Reject

**Comment:**

This paper presents work on a dataset for object concept learning.  The main contributions include causal relations in the dataset and a method (OCRN).  The initial reviews pointed to concerns over the nature of the causal relations, the presentation of the paper, and the OCRN method and its motivations / use of the do-operator.  The reviewers engaged in significant discussions after considering the authors' responses and the others' reviews.  After this delibration, the concerns over the dataset, its annotations, and the presentation of the methods were deemed to be better served by a full revision and reconsideration of the paper.  As such, the paper is not recommended for acceptance at this time.